# 🎓 AAAR-1.0: Assessing AI's Potential to Assist Research

**Renze Lou** [1]  **Hanzi Xu** [2]  **Sijia Wang** [3]  **Jiangshu Du** [4]  **Ryo Kamoi** [1]  **Xiaoxin Lu** [1]  **Jian Xie** [5]  **Yuxuan Sun** [5]
**Yusen Zhang** [1]  **Jihyun Janice Ahn** [1]  **Hongchao Fang** [1]  **Zhuoyang Zou** [1]  **Wenchao Ma** [1]  **Xi Li** [6]  **Kai Zhang** [7]
**Congying Xia** [5]  **Lifu Huang** [3]  **Wenpeng Yin** [1]

## Abstract

Numerous studies have assessed the proficiency of AI systems, particularly large language models (LLMs), in facilitating everyday tasks such as email writing, question answering, and creative content generation. However, researchers face unique challenges and opportunities in leveraging LLMs for their own work, such as brainstorming research ideas, designing experiments, and writing or reviewing papers. In this study, we introduce AAAR-1.0, a benchmark dataset designed to evaluate LLM performance in three fundamental, expertise-intensive research tasks: (i) EQUATIONINFERENCE, assessing the correctness of equations based on the contextual information in paper submissions; (ii) EXPERIMENT-DESIGN, designing experiments to validate research ideas and solutions; and (iii) PAPERWEAK-NESS, identifying weaknesses in paper submissions. AAAR-1.0 differs from prior benchmarks in two key ways: first, it is explicitly research-oriented, with tasks requiring deep domain expertise; second, it is researcher-oriented, mirroring the primary activities that researchers engage in on a daily basis. An evaluation of both open-source and closed-source LLMs reveals their potential as well as limitations in conducting sophisticated research tasks. We will keep iterating AAAR-1.0 to new versions. Project Webpage: https://renzelou.github.io/AAAR-1.0/

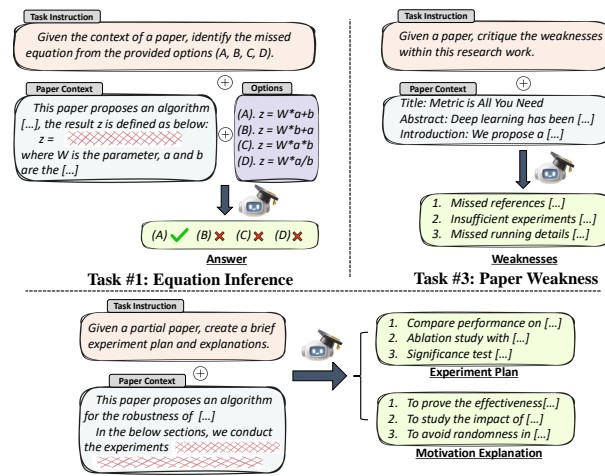

Figure 1: The input-output illustration of three tasks in the proposed AAAR-1.0 benchmark.

## 1. Introduction

Although AI has brought transformative changes to various aspects of life, its impact on researchers unfolds in a nuanced manner. On the one hand, AI assists in various research disciplines, such as Social Science (Neuman et al., 2023), Finance (Gu et al., 2024), Medicine (Rakhimov et al., 2022), GeoScience (Praskievicz, 2018), etc., significantly expediting academic processes. However, many of these applications are superficial, often limited to data-driven clustering or classification. On the flip side, the AI era poses challenges for researchers. Despite its ability to streamline some activities, researchers still face demanding, cognitively intensive tasks such as staying current through extensive paper reading, rapidly generating ideas in response to fast-paced advancements, conducting rigorous experiments to substantiate claims, and managing an increasing volume of peer reviews. Then a question looms: *How effectively can AI assist researchers in tasks that are domain-specific, expertise-demanding, and reasoning-intensive?*

Existing works proved the promising potential for using LLMs in assisting AI research. Si et al. (2024) conducted a large-scale human study and found that LLMs can gen-

Work done prior to Jiangshu joining Amazon. [1]Pennsylvania State University; [2]Netflix; [3]University of California, Davis; [4]University of Illinois Chicago; [5]Individual Researcher; [6]University of Alabama at Birmingham; [7]Ohio State University. Correspondence to: Renze Lou <renze.lou@psu.edu>, Wenpeng Yin <wenpeng@psu.edu>.

*Proceedings of the 42nd International Conference on Machine Learning*, Vancouver, Canada. PMLR 267, 2025. Copyright 2025 by the author(s).

erate creative research ideas. Lu et al. (2024) proposed an autonomous agent to handle complicated research workflow and write a whole research paper. However, most of these works focus on addressing highly subjective problems that require a high degree of expertise, making evaluation laborious and hard to reproduce. This underscores the need for a comprehensive benchmark that rigorously assesses LLMs' capabilities in expertise-intensive research activities.

To this end, in this work, we introduce AAAR-1.0, a novel benchmark that aims to comprehensively assess the LLMs' capacity on expert-level research tasks. As illustrated in Figure 1, AAAR-1.0 decomposes three distinct expert-level AI research tasks from the researcher's daily activities, including i) EQUATIONINFERENCE, investigating whether the LLMs can infer the equation correctness based on the paper context; ii) EXPERIMENTDESIGN, validating LLMs' ability on designing reliable experiments for a research idea; and iii) PAPERWEAKNESS, testing the quality of weaknesses discovered by LLMs from paper drafts. To ensure data quality, senior AI researchers with extensive domain expertise perform data annotation for AAAR-1.0, followed by rigorous multi-round data examination and filtering. All three tasks require models to possess strong domain knowledge covering various cutting-edge research findings, as well as expert-level research experience, to the extent that even humans need substantial research accumulation to tackle the tasks we designed. Crucially, tasks here are singular, standalone challenges (with clear input and output expectations) rather than a complicated task chain (Li et al., 2024; Lu et al., 2024), providing a more transparent assessment of the model's intermediate output. Benefiting from the proposed automatic metrics, we conduct extensive experiments across numerous mainstream LLMs, where we find that:

- With a random guess baseline of 40% $F_1$, the performance of most LLMs on EQINFER hovers just slightly above chance, with the top models reaching around 46%. This highlights the difficulty of the task, despite its reliance primarily on local context reasoning.

- In EXPDESIGN, LLM-designed experiments are innovative and more diverse than those by humans; however, many are trivial, lack feasibility, and stray from the original research objectives.

- In PAPERWEAKNESS, LLM-identified weaknesses often lack depth and specificity, making them broadly applicable and less useful for providing feedback on paper drafts.

## 2. Related Work

**LLMs for AI Research.** With the rapid evolution of pertaining techniques, LLMs are found to be useful in assisting various research disciplines (Yu et al., 2024a; Labrak et al.,

2024), particularly in AI research, such as generating novel research ideas (Kumar et al., 2024; Yu et al., 2024b), reviewing research draft (Gao et al., 2024; Du et al., 2024; Liang et al., 2024; Zhu et al., 2025), and writing scientific papers (Chamoun et al., 2024; Lu et al., 2024; Weng et al., 2024). For example, Si et al. (2024) conducted a large-scale human investigation on LLM-generated research ideas and found that LLMs can generate novel ideas compared with humans while lacking feasibility. Du et al. (2024) found that while LLMs are effective at summarizing papers, they tend to overly trust the authors' claimed strengths and struggle to identify weaknesses specific to the paper. Furthermore, some works try to employ LLMs to solve more complicated research tasks that are composed of multiple steps (Li et al., 2024; 2023; Tang et al., 2023). Notably, Lu et al. (2024) proposed AI-SCIENTIST, an autonomous agent framework that can handle a series of challenging research tasks consecutively, including generating research ideas, coming up with the corresponding experiments along with the implementations, and then writing the final research paper — exactly how human conduct a whole research pipeline. However, there is still a lack of systematic evaluations and quantitative analyses on the LLMs' (intermediate) output of each single-step research task. Accordingly, our work focuses on building a benchmark consisting of individual research steps with clear input-output expectations, making it suitable for comprehensive LLM evaluation. Moreover, **we emphasize that relying on LLMs to fully replace human effort might compromise academic integrity**. While our benchmark primarily serves an educational purpose — LLMs assist junior researchers by providing imperfect but insightful ideas, rather than by governing the entire research process.

**Benchmarks for AI Research Tasks.** Existing "LLM assists research" benchmarks mainly focus on the implementation and execution part of the research pipeline (Lu et al., 2024; Chen et al., 2024a; Li et al., 2024; Chan et al., 2024). For instance, Huang et al. (2024) proposed MLAgentBench to test the LLMs' capacity for writing project code and training the ML models, where the evaluation metric is the test performance of the models trained by LLMs. However, realworld AI research activities are diverse and some of them are hard to assess for quality, such as generating research ideas, which requires intensive manual assessment (Si et al., 2024; Liang et al., 2024). Our work centers on tasks that emphasize a comprehensive mastery of the scientific research field and core elements of a researcher's daily workload, and we try to build curated task-specific metrics for every single task for a more efficient and accurate LLMs appraisal.

## 3. AAAR-1.0

Figure 2 provides a data construction overview. In the following sections, we elaborate on the data collection de-

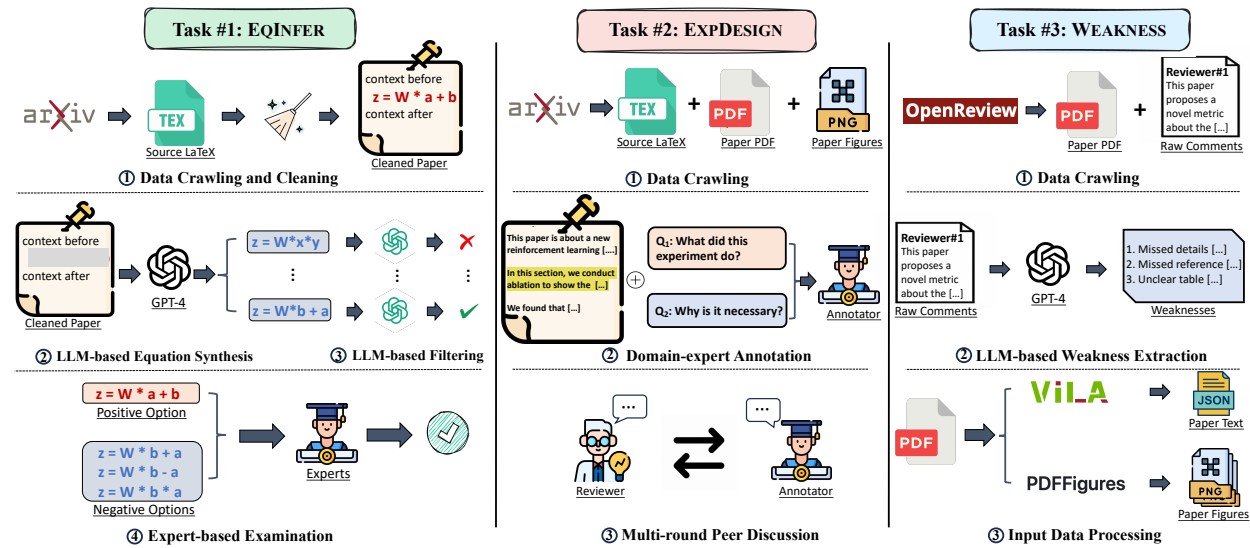

Figure 2: Data construction workflows of the three tasks in 🎥 AAAR-1.0.

tails, including § 3.1 EQUATIONINFERENCE ( **EQINFER** ), § 3.2 EXPERIMENTDESIGN ( **EXPDESIGN** ), and § 3.3 PA-PERWEAKNESS ( **WEAKNESS** ).

### 3.1. EQUATIONINFERENCE

Crafting a correct scientific equation in paper writing or validating an equation in paper reviewing is challenging, as it requires a thorough understanding of an algorithm or the intricate relationships among numerous variables. Directly prompting LLMs to generate equations proves overly demanding. Therefore, this work formulates **EQINFER** (Figure 1) as a binary inference task.[1]

① **Data crawling and cleaning.** For the data source, we adopt the pre-compilation LaTeX code for two reasons: i) existing PDF parsing tools, such as PyMuPDF and Paper-Mage (Lo et al., 2023), can introduce considerable noise to the parsed equation text; ii) considering most of exiting LLMs are capable with processing LaTeX code, using LaTeX source instead of parsed text can be more accurate and provide LLMs with richer information. Meanwhile, we only crawl those peer-reviewed papers accepted by top-tier conferences to avoid using low-quality human-written equations. Accordingly, we first obtain the accepted paper list from ACL Anthology, from year 2019 to 2023. Next, we search each paper on arXiv to crawl its LaTeX source (if it exists). Finally, we get a total of 1,762 papers' source LaTeX packages. We then clean the LaTeX sources by deleting all the comments and combining multiple cross-referred .tex files into a main file. Afterward, we use regex to randomly extract (at most) 3 equations' code snippets per paper, resulting in 3,877 human-written equations.

② **LLM-based equation synthesis.** As EQINFER assessing whether the LLMs can infer the correctness of equation (i.e., binary classification), for each human-written positive equation, we have to craft counterpart negative equations. To this end, for each positive equation, we prompt GPT-4 to synthesize a negative equation based on the paper context. We repeat this prompt (with a high decoding temperature) until three different negative equations are synthesized.[2]

③ **LLM-based filtering.** However, the LLM-synthetic equations can be context-unaligned, i.e., some synthesized equations contain notation that is never defined in the paper context, which becomes a superficial shortcut and too effortless for LLMs to identify. To improve data quality, we prompt GPT-4 to identify context-unaligned negative equations. We then eliminate the positive equation and its negative counterparts, where all three negative counterparts are unaligned. This filtering leads to a final of 1,449 positive equations and 4,347 negative equations (each positive equation has three negative counterparts, and at least one negative counterpart is "challenging").

④ **Expert-based examination.** Furthermore, it's also possible that synthesized negative equations are actually correct (i.e., false negative) — even if the negative and positive equations are written differently, the final compiled results might be the same. We then employ human experts to review the data further and filter out false negative equations, checking the classification instances for accuracy.

We asked 5 senior PhD students who are experienced in AI research to check all instances. We ask human experts to consider the following criteria for each positive equation and its negative counterparts (each pair): i) **Are all equations**

---

[1] EQINFER also facilitates a multiple-choice QA setting; while we find a binary inference is more challenging for LLMs.

[2] The number of negative equations is empirically decided.

**grammatically correct?** ii) **After compilation, are all negative equations different from the positive ones?** We ask every human expert to use external LaTeX compilation tools (e.g., TeXlive), and identify the pairs that cannot meet the criteria. Each pair is examined by at least two experts, and we only keep pairs that all experts decide to keep. After this strict examination, a total of 1,049 pairs are eventually kept (27.6% pairs are filtered)

**Final data.** We finally obtain 1,049 positive equations (each has three negative counterparts). We show data statistics of EQINFER in Table 7 and data examples in Figure 8.

## 3.2. EXPERIMENTDESIGN

Given a research topic, such as a novel ML algorithm, a qualified researcher can design a solid experiment plan for it, and clarify underlying motivation to ensure the reliability of the designed experiment. Unlike the concurrent works that focus on the experiment implementation (Lu et al., 2024; Huang et al., 2024), we emphasize the importance of assessing the high-level experiment design of LLMs before the subsequent implementation to avoid any expensive execution iteration. Therefore, as shown in Figure 1, we formulate **EXPDESIGN** as a text-generation task that takes pre-experiment paper context as input, and then generates the experiment and explanation list.

① **Data crawling.** As for the data source, we first collect $\geq$ 10k papers' data from arXiv, including LaTeX sources and PDFs, which cover broad AI categories, including `cs.AI`, `cs.CL`, and `cs.CV`, from year 2018 to 2023. Similarly, to ensure the source data quality, we only use papers that have appeared at well-known conferences.

② **Domain-expert annotation.** Making a reliable and executable experiment plan requires solid foundation knowledge of a specific research area. Consequently, we set a high standard for choosing annotators: i) be a senior Ph.D. student with at least one peer-reviewed publication in leading AI venues; ii) have more than 4 years of AI research experience; iii) frequently serve as conference reviewers. Finally, we invite a total of 10 qualified experts to participate in our data collection procedure. Given the 10k crawled papers, we first ask every annotator to bid on the papers that they are interested in. After bidding, each of them is assigned 10 papers, i.e., a total of 100 papers to be annotated. During annotation, we post each paper PDF on online Google Drive and ask the annotator to first carefully read the whole paper. Then, we ask them to identify and locate the key experiments in each paper (i.e., highlighting the relevant paragraphs of each experiment). We don't consider some trivial experiments, such as those supplemental analyses in the appendix section. For each identified experiment, the annotator has to concisely answer two questions: i) **What did this experiment do?** ii) **Why did the paper authors conduct this experiment?** In other words, we ask the annotator to summarize all the key experiments in this paper and explain the underlying motivations based on their rich domain experience.

③ **Multi-round peer discussion.** Intuitively, different experts might have different opinions on the same research topic. Particularly, when explaining the underlying motivation of an experiment, adopting only a single expert's opinion might introduce bias to our annotation. Hence, we conduct a further multi-round peer discussion. For each paper, where all the key experiments are identified, summarized, and explained, we ask a different expert (reviewer) to review the annotation by considering the following three criteria: i) **Are the identified experiments all the key experiments?** ii) **Does each experiment summarization covers all key information?** iii) **Does each explanation sound reasonable and reliable?** Each reviewer must leave comments on the online PDF regarding the above criteria, and then the annotator must respond to each comment — either accept the suggestion and revise the previous annotation or provide a "rebuttal" to the reviewer to uphold the annotation. This discussion is iterative until both opinions align. Eventually, for each paper, we collect two lists: i) the experiment list, summarizing each experiment step of the paper; ii) the explanation list, the underlying motivations that are one-one corresponding to the experiment.

**Final data.** After annotation, we use the pre-experiment context of each paper (according to the first-experiment location identified by the annotator) as the input. Furthermore, we use GPT-4 to delete any sentence that potentially leaks the experiment from the input.[3] Similar to the EQINFER, we utilize the source LaTeX as the input text to avoid PDF paring noise. As for the image input, we collect those figures within each paper's source LaTeX package and only keep figures that are used in the pre-experiment context. Overall, a total of 100 instances are collected. As shown in Figure 1, the input of each instance is the pre-experiment context (including the figures), and the ground-truth output is the expert-annotated experiment plan and the explanations. Table 8 shows data statistics and Figure 9 illustrates the sample case in EXPDESIGN.

## 3.3. PAPERWEAKNESS

Another critical research task is paper review. Previous works have demonstrated the usefulness of the LLM-based review feedback (Gao et al., 2024; Jin et al., 2024; Lu et al., 2024). However, as indicated by Du et al. (2024); Liang et al. (2024), LLMs only excel at summarizing the research

---

[3]About 9.8% sentences are deleted.

strengths while falling significantly short on weakness criticism. Hence, we build **WEAKNESS** for particularly investigating the LLM-generated weaknesses.

① **Data crawling.** We first crawl a total of 3,779 anonymous submissions of *ICLR 2023* from OpenReview,[4] including PDF and other meta information (e.g., scores, decisions, and tracks). As the *ICLR 2023* has 13 distinct tracks while the paper distribution across different tracks is highly biased, we then uniformly sample papers from different research tracks to improve the domain diversity. Meanwhile, during sampling, we also keep the accept/reject papers distributed equally to avoid data bias. In a word, we finally collect a total of 1,000 papers (500 accepted; 500 rejected), uniformly covering all 13 tracks. Please refer to Figure 3 for the track and score distribution of the 1,000 papers.

② **Extraction of human-written weaknesses.** Since the raw comments crawled from *ICLR 2023* are mixed with both strengths and weaknesses, we further employ GPT-4 to extract all the weaknesses from each reviewer's comments and compose multiple weaknesses into a list. Notably, we force GPT-4 to keep the original text of the reviewer, i.e., all weaknesses in our dataset are those original sentences written by the reviewer without any modifications.[5] What's more, sometimes one reviewer might repeatedly mention the same weakness throughout the comment. In this case, we simply keep all the repeated weaknesses because, if one weakness is repeatedly mentioned by the reviewer, it's intuitively an important weakness that the reviewer wants to emphasise; accordingly, keeping the repeat items can penalize LLMs more on missing this weakness.

For each paper, we can finally get multiple weakness lists (one weakness list per reviewer, one paper can have multiple reviewers). We further delete a few papers without any weaknesses found in the raw comments, resulting in a total of 993 instances, i.e., 993 {paper, weakness lists} pairs.

③ **Input data processing.** As we mentioned before, we crawl papers from OpenReview instead of arXiv because the under-review paper draft is required for this task. However, not every paper from OpenReview can be found on arXiv, i.e., the source LaTeX code and figures of most under-review papers are unavailable. Therefore, we utilize VILA (Lin et al., 2023) to parse text data out from the PDF; we also employ PDFFigures-2.0 (Clark & Divvala, 2016) to extract all the figures and tables (in image) from the paper, as Vila is not good at processing the table data.

---

[4]We adopt ICLR because it releases full submissions, while some other conferences only release accepted papers.

[5]We manually checked GPT-4's extraction results of 200 cases — GPT-4 only missed ≤1% of reviewer-written weaknesses and maintained almost all the original text.

**Final data.** Our final data is composed of 993 instances, each input is paper text along with figure/table images, and each output is peer reviewers' weakness lists. Table 9 shows data statistics; Figure 10 presents an example of the data instances. We show the data diversity (score and track distribution) in Figure 3.

## 4. Evaluation Criteria

For **EQINFER** , we adopt $F_1$ as the classification criterion. For EXPDESIGN and WEAKNESS, since both tasks have free-form outputs, we develop several novel task-specific metrics in addition to the conventional ROUGE (Lin, 2004).

We use LLMs to evaluate the experiment list of **EXPDESIGN** . Specifically, given a model-predicted experiment list $p$, and the ground-truth list $g$, we calculate:

$$\text{En-Precision} = \frac{1}{m} \sum_{i=1}^{m} f(p_i, g) \qquad (1)$$

$$\text{En-Recall} = \frac{1}{n} \sum_{j=1}^{n} f(g_j, p) \qquad (2)$$

where the $m$ and $n$ are the list length of $p$ and $g$; $f(.)$ represents the LLM prompting, where we prompt LLM to decide whether each predicted experiment item ($p_i$) is entailed by the whole ground-truth list ($g$), proceeding with binary output, and vice versa. Intuitively, En-Precision reflects how many prediction experiments match ground-truth experiments. In this work, we used GPT-4o as an evaluator.

While for the explanation generation of EXPDESIGN, as the prediction experiments are one-on-one corresponding to the ground truth, we adopt a semantic-based metric:

$$\text{S-Match} = \frac{1}{m} \sum_{i=1}^{m} \text{sim}(p_i, g_i) \qquad (3)$$

where we use SentenceBERT (Reimers, 2019) to measure the semantic similarity between $p_i$ and $g_j$.

Unlike EXPDESIGN, the ground truth of **WEAKNESS** is multiple reviewers' weakness lists. Instead of merely merging the opinions of various reviewers into one flattened list and keeping LLM-as-judge as the metric (which is not only costly but also loses the structural information of diverse research perspectives), we employ the following semantic-based metric to efficiently evaluate predicted weaknesses:

$$\text{S-Precision} = \frac{1}{m} \sum_{i=1}^{m} \left( \frac{1}{r} \sum_{k=1}^{r} \max_{j} \text{sim}(p_i, g_j^k) \right) \qquad (4)$$

$$\text{S-Recall} = \frac{1}{r} \sum_{k=1}^{r} \left( \frac{1}{n_k} \sum_{j=1}^{n_k} \max_{i} \text{sim}(g_j^k, p_i) \right) \qquad (5)$$

where $r$ is the number of reviewers of the given paper, $n_k$ means the length of $k$-th reviewer's weakness list, and $g_j^k$

indicates the $j$-th item in $k$-th reviewer's weakness list.

Additionally, in the real world, we would think a review weakness is reliable if it is specific to a paper. Meanwhile, we also hope the review is informative, i.e., no excessive similar weaknesses in one review. Inspired by the classic TF-IDF, we propose a novel review diversity metric:

$$\text{ITF-IDF} = \frac{1}{w} \sum_{j=1}^{w} \left( \frac{1}{m_j} \sum_{i=1}^{m_j} \log\left(\frac{m_j}{O_i^j}\right) \times \log\left(\frac{w}{R_i^j}\right) \right) \quad (6)$$

$$O_i^j = \sum_{k=1}^{m_j} \text{sim}(p_i^j, p_k^j) \quad (7)$$

$$R_i^j = \sum_{l=1}^{w} \max_s \text{sim}(p_i^j, p_s^l) \quad (8)$$

where the $w$ is the total number of papers in the dataset, $p^j$ is $j$-th paper's prediction weakness list, $p_i^j$ is the $i$-th weakness in $p^j$. Moreover, $O_i^j$ calculates the intra-paper occurrence frequency of $p_i^j$; $R_i^j$ is the "soft" number of papers that also contain the $p_i^j$, which is computed by summing the maximum similarity scores between $p_i^j$ and other paper's weaknesses. In a word, $O_i^j$ measures informativeness, and $R_i^j$ measures specificity. The complete ITF-IDF consider both aspects and reflects the overall weakness diversity.

## 5. Experiments and Analyses

In this section, we conduct extensive experiments on AAAR-1.0, across various mainstream LLMs, to quantify the current LLMs' capacity to tackle high-level research tasks. Specifically, § 5.1 for EQINFER, § 5.2 for EXPDESIGN, and § 5.3 for WEAKNESS. Please refer to Appendix B.2 for running details of the LLMs.

### 5.1. EQUATIONINFERENCE

**Settings.** As different LLMs have distinct context windows, to ensure a fair comparison, we fix the maximum input length for all models. According to Table 7, we empirically use 1,000 words for both contexts before and after equations, i.e., 2,000 surrounding words.

**Main results.** Table 1 shows the main results. Firstly, a simple baseline that predicts all equations as positive achieves 40% $F_1$ (due to the 1:3 of positive and negative equations), while nearly all open-source LLMs even cannot beat this naive baseline. Notably, though the performance of Mixtral is slightly superior to the baseline, the extremely biased precision and recall scores imply that Mixtral is also simply predicting almost all samples as positive instead of truly inferring. Meanwhile, compared to the All-Positive baseline, the performance superiority of the strong close-source LLMs is not significant, the best LLM on this task only obtains 47.98%, which demonstrates the challenge of EQINFER compared with other similar benchmarks (Song

Table 1: Various LLMs' performances on EQINFER task (1,049 positive and 3,147 negative samples). "All-positive" indicates a baseline that predicts all equations as positive.

| Methods | $F_1$ | Prec. | Rec. |
|---|---|---|---|
| All-Positive | 40.00 | 25.00 | 100.00 |
| *Open-source LLMs* | | | |
| OLMo-7B (Groeneveld et al., 2024) | 13.64 | 11.93 | 15.91 |
| Mistral-7B (Jiang et al., 2023) | 28.45 | 19.28 | 54.24 |
| Mixtral-8x22B-MoE (Jiang et al., 2024) | 40.90 | 26.15 | 93.80 |
| Qwen 2.5-72B (Qwen Team, 2024) | 31.22 | 26.28 | 57.40 |
| Llama 3.1-70B (MetaAI, 2024) | 33.08 | 22.14 | 65.39 |
| *Closed-source LLMs* | | | |
| Gemini 1.5 Pro (Anil et al., 2023) | 46.74 | 32.05 | 86.27 |
| Claude 3.5 sonnet (Anthropic, 2024) | 45.13 | 29.48 | **96.18** |
| GPT-4o (OpenAI, 2024a) | 40.35 | 30.79 | 58.53 |
| o1-preview (OpenAI, 2024b) | 46.35 | 31.43 | 88.27 |
| o3-mini (OpenAI, 2025) | **47.98** | **34.34** | 79.59 |

et al., 2023). The generally high recall with low precision of all LLMs also indicates real-world risks, e.g., relying on LLMs to check the validity of equations in paper review.

$\mathcal{Q}$: **Do more contexts boost performance?** EQINFER places high demands on reasoning within the scientific context. To quantify the impact of input context length, we scale the input length (per side) from 100 to 1,500 words. As shown in Figure 4, for the open-source LLMs (Llama and Qwen), an appropriate context length can boost the performance; while for GPT-4o, scaling up the context length doesn't contribute much to the $F_1$. However, during the scaling, we find that the precision of GPT-4o is gradually increased, and the recall is decreased accordingly; considering the label distribution of EQINFER, we believe precision can better reflect the model's true capacities on this task. Thus, we anticipate that scaling up context shall be beneficial to those strong close-source LLMs such as GPT-4o.

### 5.2. EXPERIMENTDESIGN

**Settings.** Similarly, we unify the input context length of different LLMs to ensure a fair comparison. According to Table 8, we set 2,000 and 3,000 input words for open- and closed-source LLMs, respectively. Meanwhile, as experiment explanation is the subsequent task of experiment design, using model-generated experiments can propagate errors in explanation, leading to inferior results for most LLMs. To this end, we provide LLMs with the oracle experiments when generating explanations.

**Main results.** Table 2 shows the main results. For the experiment design, the closed-source LLMs generally outperform open-source LLMs. However, the score values of all LLMs are relatively low (20%~30%), implying the LLMs consistently miss ground-truth experiments from the origin paper (low recall), and they tend to generate

Table 2: Various LLMs' performances on the 100 instances of EXPDESIGN. The explanation generation is based on the oracle experiments to prevent error propagation. "Copy Input" directly copies each experiment idea as the explanation.

| Methods | Experiment Design | | | Experiment Explanation | | |
|---|---|---|---|---|---|---|
| | En-F$_1$ | En-Precision | En-Recall | S-Match | ROUGE-L | ROUGE-1 |
| Copy Input | — | — | — | 40.32 | 22.06 | 25.28 |
| *Open-source LLMs* | | | | | | |
| OLMo-7B (Groeneveld et al., 2024) | 14.80 | 17.50 | 19.80 | 45.78 | 26.30 | 30.38 |
| Mistral-7B (Jiang et al., 2023) | 18.96 | 24.83 | 21.38 | 50.18 | **30.20** | 34.69 |
| Mixtral-8x22B-MoE (Jiang et al., 2024) | 23.16 | 24.45 | 30.57 | 49.07 | 29.96 | 34.53 |
| Llama 3.1-70B (MetaAI, 2024) | 22.92 | 23.10 | 29.76 | 50.05 | 29.33 | 34.11 |
| Qwen 2.5-72B (Qwen Team, 2024) | 24.28 | 22.48 | 34.44 | 51.12 | 29.46 | 34.68 |
| *Closed-source LLMs* | | | | | | |
| Gemini 1.5 Pro (Anil et al., 2023) | 27.25 | 28.66 | 34.92 | 52.87 | 28.52 | 33.80 |
| Claude 3.5 sonnet (Anthropic, 2024) | 27.99 | 24.48 | **42.09** | 53.03 | 18.75 | 26.15 |
| GPT-4o (OpenAI, 2024a) | 25.03 | 22.25 | 36.59 | 54.79 | 27.54 | 34.31 |
| o1-preview (OpenAI, 2024b) | 30.13 | 28.13 | 38.59 | **58.55** | 29.11 | **36.70** |
| o3-mini (OpenAI, 2025) | **30.17** | **28.70** | 37.67 | 54.01 | 20.71 | 29.14 |

more novel experiments that didn't show in the origin paper (low precision). As for the experiment explanation, the S-Match scores of closed-source LLMs still surpass the open-source LLMs. Furthermore, there is a negative correlation between S-Match and ROUGE score, where the ROUGE scores of closed-source LLMs are broadly inferior. We find that the open-source LLMs often try to copy the terms or phrases from the given experiment, or even simply paraphrase the experiment instead of explaining, which results in a high superficial overlap with the ground-truth explanation. This observation highlights the importance of adopting the proposed S-Match to avoid evaluation bias of traditional generation metrics.

$\mathcal{Q}_1$: **What is the quality of the model-generated novel experiments?** The low En-Precision of LLMs in Table 2 indicates the creativity of LLMs in generating novel experiments. We then randomly sample 15 papers from the EXPDESIGN and ask 3 experts to manually review the model-generated novel experiments. Specifically, we ask the experts to judge the necessity of the novel experiments, where we set three necessity levels: "A" indicates the experiment is necessary/mandatory to support the main claim, "B" represents optional/supplementary experiments, and "C" for those unrelated experiments (see Appendix C.2 for evaluation details). Table 3 shows the necessity scores of the three strongest LLMs. We find that LLMs consistently generate a lot of novel experiments, especially the Claude; though most of them are optional, even fancy/unrelated experiments, there are still a considerable amount of necessary experiments generated, e.g., the results of o1. We further find that some novel experiments can be regarded as useful supplementary analyses w.r.t. the human experiments. Table 11 shows examples of model-suggested experiments.

Table 3: The human evaluation results on the novel experiments suggested by LLMs. "A", "B", and "C" represent the different quality level (i.e., necessity); "A" is the best level.

| Models | # of novel EXP | Necessity (%) | |
|---|---|---|---|
| | | A | B |
| Gemini 1.5 Pro | 59 | 30.59 | 45.76 |
| Claude 3.5 sonnet | 112 | 21.78 | 50.00 |
| o1-preview | 71 | 35.84 | 36.61 |

Table 4: The impact on S-Match scores of maintaining the experiment's self-containment for EXPDESIGN.

| Models | One-by-One | Whole-List |
|---|---|---|
| Llama 3.1-70B | 50.05 | 49.36 (↓ 0.7) |
| Qwen 2.5-72B | 51.12 | 48.56 (↓ 2.6) |
| Gemini 1.5 Pro | 52.87 | 57.48 (↑ 4.6) |
| Claude 3.5 sonnet | 53.03 | 59.11 (↑ 6.1) |
| GPT-4 | 55.03 | 56.95 (↑ 1.9) |
| GPT-4o | 54.79 | 58.54 (↑ 3.8) |
| o1-preview | 58.55 | 61.58 (↑ 3.0) |

$\mathcal{Q}_2$: **Can self-contained experiment design enhance the experiment explanation?** When generating the explanation in Table 2, we provide LLMs with each individual experiment and let them explain one by one, because we find that, when providing the whole experiment list, those open-source models only explain partial experiments because of their poor instruction-following capacity. However, there are intuitively some semantic or logical relations between different experiments, e.g., some experiments are prerequi-

Table 5: The human evaluation results on LLMs' output explanations of ExpDesign . "Acc. ratio" means how many model outputs are accepted by the annotator.

| Models | Acc. ratio |
|---|---|
| Llama 3.1-70B | 22.93 |
| Gemini 1.5 Pro | 55.07 |
| Claude 3.5 sonnet | 61.46 |
| GPT-4o | 69.72 |
| o1-preview | **76.14** |

sites to others. Therefore, this one-by-one prompting might break the self-containment of an experiment plan. Consequently, we test with the "whole-list" prompting, where the LLMs are given the complete experiment list and are asked to explain all experiment steps together.

As shown in Table 4, unlike the open-source LLMs, the explanation performances of those closed-source LLMs are generally improved after adopting whole-list prompting. According to further manual checking, after maintaining the self-containment of the experiments, the LLMs can refer to other experiments and better grasp the underlying motivation of the current experiment.

$\mathcal{Q}_3$: **Do human evaluation results align with automatic metrics for explanation?**    As the explanation can be open-ended, in this paragraph, we provide the human evaluation results on different LLMs' experiment explanation outputs. In detail, we randomly select 20 out of 100 papers and ask 5 annotators to read the experiments along with each model's explanations; we then let the annotator decide whether each model's explanation is acceptable (see Appendix C.3 for more details). Table 5 illustrates the results, where the score variance is higher than Table 2. However, the performance ranking of both tables is perfectly correlated with each other (Spearman's rank correlation coefficient = 1), demonstrating the effectiveness of S-Match.

$\mathcal{Q}_4$: **Do more contexts boost performance?**    We also investigate the impact of input context length for ExpDesign. As shown in Figure 5, we scale up the input pre-experiment context length from 0.1k to 10k tokens (10k is the length of the longest paper). For the experiment design, more input context does improve the performance of different LLMs, while this benefit stops after exceeding 8k tokens, which means that after the necessary information has been covered, scaling context becomes inefficient. Meanwhile, the explanation generation results reveal that LLMs primarily depend on given experiments rather than paper context to explain motivations. However, we do not expect this as we hope LLMs can explain the motivation based on a thorough

understanding of the paper, just like how human experts do. Hence, there is still a considerable gap between the LLMs and humans in terms of grasping research motivations.

$\mathcal{Q}_5$: **Does multi-modal input boost performance?**    Intuitively, besides the text, when designing experiments for a given research topic, the figures can provide rich supplementary information, such as an algorithm illustration that can help better understand this research topic and underlying motivations. Hence, we test the performance of different LMMs (Large Multimodal Models), including GPT4-o and InternVL2 (Chen et al., 2024b). Table 12 shows the ablation results on the figure data. To our surprise, the figure data doesn't improve the LMMs' results in this task, even harming the performances. This might be due to the low informativeness of the figures, as figures usually consume more input tokens but act only as supplementary information to the text, indicating future work on developing LMMs that can effectively leverage the scientific figures.

### 5.3. PaperWeakness

**Settings.**    Intuitively, full paper content is necessary for paper reviewing. Therefore, instead of setting a maximum input length, in Weakness, we try to utilize the whole paper. As the input length of Weakness is extremely long (see Table 9), we adopt a "split-combine" method — we first split the whole paper into smaller pieces and let LLMs predict the weaknesses of each piece separately; after that, we merge all pieces' weaknesses as a final prediction. For the length of each small piece, we set 2,000 and 3,000 words for open- and closed-source LLMs, respectively. Additionally, in this task, we also examine the performance of AI-SCI (Lu et al., 2024), which enhances LLMs' paper review ability by leveraging advanced prompting techniques, e.g., self-reflection (Shinn et al., 2024) and response ensembling (Wang et al., 2023).[6]

**Main results.**    Table 6 shows the main results, where the closed-source LLMs' overall performances are generally superior to the results of open-source LLMs. Similarly, closed-source LLMs are particularly excellent in S-Recall because of more generated weaknesses. However, there is still a considerable gap in the weakness diversity between the LLMs and human experts.[7] Compared with human review, most LLM-generated weaknesses are vague and lack the necessary knowledge about some frontier research works. Surprisingly, AI-SCI performs worse than backbone GPT-

---

[6]We don't run AI-SCI on ExpDesign, because AI-SCI takes model-generated ideas as the inputs, which are incompatible with our task setting.

[7]The human's ITF-IDF score in Table 6 can be slightly underestimated. This is because we keep the repeated weaknesses in the human review, which affects the human review's informativeness (lower ITF) but is useful when calculating the S-Recall for LLMs.

Table 6: Various LLMs' performances on the 993 instances of WEAKNESS.

| Methods | S-$F_1$ (%) | S-Precision (%) | S-Recall (%) | Weakness Diversity |
|---|---|---|---|---|
| | | | | ITF-IDF (↑) |
| Human Review | — | — | — | 7.69 |
| *Open-source LLMs* | | | | |
| OLMo-7B (Groeneveld et al., 2024) | 43.25 | 40.38 | 47.04 | 2.45 |
| Mistral-7B (Jiang et al., 2023) | 42.03 | 43.80 | 40.77 | 1.17 |
| Mixtral-8x22B-MoE (Jiang et al., 2024) | 43.23 | **44.59** | 42.23 | 0.98 |
| Llama 3.1-70B (MetaAI, 2024) | 42.78 | 43.19 | 42.70 | 2.60 |
| Qwen 2.5-72B (Qwen Team, 2024) | 42.74 | 43.80 | 42.05 | 1.21 |
| *Closed-source LLMs* | | | | |
| Gemini 1.5 Pro (Anil et al., 2023) | **48.75** | 43.97 | 55.08 | 5.88 |
| Claude 3.5 sonnet (Anthropic, 2024) | 47.85 | 41.97 | 56.00 | 3.91 |
| GPT-4o (OpenAI, 2024a) | 47.73 | 42.09 | 55.48 | **5.95** |
| o1-preview (OpenAI, 2024b) | 48.62 | 42.54 | **57.08** | 5.63 |
| o3-mini (OpenAI, 2025) | 46.33 | 42.00 | 51.99 | 5.85 |
| *LLM Agent Framework* | | | | |
| AI-SCI (GPT-4o) (Lu et al., 2024) | 45.05 | 40.02 | 51.91 | 2.23 |

4o, especially on ITF-IDF, which suggests the challenge of WEAKNESS, i.e., simply adopting popular prompting techniques cannot well address this task.

$\mathcal{Q}_1$: **Is the split-combine effective?**  Ideally, if the LLM has a sufficient context window size, splitting the input papers for separate processing is unnecessary. Consequently, in this paragraph, we utilize the LLMs accepting long context input to compare "split-combine" with "no-split", i.e., letting LLMs write weaknesses by giving the full paper. In practice, we set the maximum number of input words to 20k, which ensures ≥95% papers in the WEAKNESS can be fully processed. As shown in Table 10, compared with giving the full paper contexts, split-combine generally brings about superior performances. During manual checking, we find that, when full paper is available, LLMs frequently neglect some important sections and omit weaknesses accordingly, while split-combine ensures that the LLMs can carefully brainstorm weaknesses within each smaller piece. Surprisingly, the LLMs' performances with full paper context can be even worse than just remaining the first 3,000 words. This implies that even the current powerful long-context LLMs still fall short when processing long scientific documents.

$\mathcal{Q}_2$: **Does multi-modal input boost performance?**  Our dataset covers both tables and figure illustrations extracted from the paper PDF as inputs. Intuitively, when reviewing a paper, both figures and tables are critical, not only for a better understanding, but also because some weaknesses are related to tables/figures.[8] Therefore, in Table 13, we adopt

---

[8]We find that there is approximately one human-written weakness related to figures or tables in each paper.

two LMMs to investigate the effectiveness of image inputs. Overall, image information, including both figures and tables, doesn't bring significant performance improvement, i.e., only InternVL2 gains a performance boost after incorporating figures; while tables slightly drop both models' results. This is probably because the LMMs cannot reason well over the information-intensive images, especially the table images.

## 6. Conclusion

In this work, we propose AAAR-1.0, a novel benchmark targeting a comprehensive evaluation of the current LLMs' capacity in assisting AI research. AAAR-1.0 consists of distinct expertise-intensive tasks along with the curated evaluation metrics. We collect high-quality data by employing senior AI researchers and conducting strict data examinations. Extensive experiments highlight the challenges and values of AAAR-1.0.

## Acknowledgments

The authors would like to thank Ibraheem Moosa and Sarkar Snigdha Sarathi Das for assisting in the data collection.

## Impact Statement

Our study explores whether LLMs can assist human researchers in AI research. We do not advocate for AI replacing human researchers. Instead, we stress that the primary responsibility for scientific research should remain with humans to prevent societal risks, with LLMs serving as tools to

enhance research efficiency. Specifically, our work analyzes the strengths and weaknesses of LLMs to ensure researchers remain judicious in their use of these tools. Our goal is to mitigate risks while maximizing the benefits offered by LLMs. We are committed to the careful distribution of data collected in our research, ensuring it is used solely for research purposes.

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

# Appendices

Within this supplementary material, we elaborate on the following aspects:

- Appendix A: Data Statistics and Diversity

- Appendix B: Implementation Details

- Appendix C: More Experiment Results and Details

- Appendix D: Data Cases and Annotation Platform Illustration

- Appendix E: Prompt Templates

## A. Data Statistics and Diversity

We provide the detailed data statistics of three datasets in our benchmark, as shown in Table 7, 8, and 9. We use the NLTK package[9] to tokenize words and count the length. When calculating the length of equations, we use the pylatexenc tool[10] to simplify the equations first.

Meanwhile, for the WEAKNESS, we also plot the review scores distribution of the papers used in the dataset, as well as the track distribution. As can be found in Figure 3, our dataset has a decent distribution, where the papers are uniformly distributed across 13 tracks, and most papers' scores ranged from 5 to 8 (i.e., most papers are weakly rejected or accepted).

Table 7: The statistics of EQINFER . Here, the "left" and "right" input context indicates the paper contexts before and after the missed equation; "pos." means the ground-truth equations (written by the source paper authors), while "neg." is the GPT4-synthetic wrong equations.

| | |
|---|---|
| # of positive equations | 1,049 |
| # of negative equations | 3,147 |
| # of source papers | 869 |
| ave. "left" input context length (in words) | 4,377 |
| ave. "right" input context length (in words) | 6,362 |
| max "left" input context length (in words) | 24,849 |
| max "right" input context length (in words) | 32,948 |
| min "left" input context length (in words) | 711 |
| min "right" input context length (in words) | 8 |
| ave. "pos." output equation length (in character) | 55 |
| ave. "neg." output equation length (in character) | 48 |
| max "pos." output equation length (in character) | 1,039 |
| max "neg." output equation length (in character) | 306 |
| min "pos." output equation length (in character) | 6 |
| min "neg." output equation length (in character) | 4 |

## B. Implementation Details

### B.1. Metric Details

When calculating the metrics, specifically for the similarity-based scores, we utilize SentenceBERT (Reimers, 2019) to encode each segment (e.g., each experiment idea in the list) into a dense vector, and then calculate the cosine similarity,[11] which takes about 1GB of memory when running on a single A100 GPU.

---

[9] https://www.nltk.org/
[10] https://github.com/phfaist/pylatexenc
[11] https://huggingface.co/sentence-transformers/all-mpnet-base-v2

Table 8: The statistics of EXPDESIGN .

| | |
|---|---|
| # of instances | 100 |
| # of source papers | 100 |
| ave. input context length (in words) | 4,288 |
| max input context length (in words) | 9,799 |
| min input context length (in words) | 698 |
| ave. # of input figures | 2.6 |
| max # of input figures | 16.0 |
| min # of input figures | 0.0 |
| ave. length of Experiment&Explanation list | 5.7 |
| ave. length per experiment (in words) | 34.3 |
| ave. length per explanation (in words) | 27.1 |
| max length of Experiment&Explanation list | 13 |
| max length per experiment (in words) | 135 |
| max length per explanation (in words) | 89 |
| min length of Experiment&Explanation list | 2 |
| min length per experiment (in words) | 9 |
| min length per explanation (in words) | 9 |

## B.2. LLMs Running Details

In our experiments, we utilize various LLMs, including both closed and open-sourced. We list the model weight sources for the open-source LLMs:

- OLMo-7B (Groeneveld et al., 2024): https://huggingface.co/allenai/OLMo-7B

- Falcon-40B (Almazrouei et al., 2023): https://huggingface.co/tiiuae/falcon-40b

- Gemma 2-27B (Gemma Team, 2024): https://huggingface.co/google/gemma-2-27b

- Mistral-7B (Jiang et al., 2023): https://huggingface.co/mistralai/Mistral-7B-Instruct-v0.3

- Mixtral-8x22B-MoE (Jiang et al., 2024) : https://huggingface.co/mistralai/Mixtral-8x22B-Instruct-v0.1

- Llama 3.1-70B (MetaAI, 2024): https://huggingface.co/meta-llama/Llama-3.1-70B

- Qwen 2.5-72B (Qwen Team, 2024): https://huggingface.co/Qwen/Qwen2.5-72B

We use VLLM to unify the inference endpoints of all the above models.[12] We use Pytorch 2.4.0 with CUDA 12.1, and use 8 NVIDIA A100 GPUs for the LLMs inference.

Meanwhile, we use the gpt-4o-2024-08-06, gpt-4-1106-preview, o1-preview-2024-09-12, gemini-1.5-pro-002, and claude-3-5-sonnet-20240620 for the closed-source LLMs. We use LiteLLM to unify the API calling for all these LLMs.[13]

Given the unstable performance of LLMs, particularly closed-source ones, we run each model thrice during our experiments, selecting the median result from these repeated runs.

---

[12]https://github.com/vllm-project/vllm
[13]https://github.com/BerriAI/litellm

Table 9: The statistics of  WEAKNESS .

| | |
|---|---|
| # of instances | 993 |
| # of source papers | 993 |
| ave. input context length (in words) | 9,811 |
| max input context length (in words) | 49,195 |
| min input context length (in words) | 24 |
| ave. # of input figures | 7.0 |
| max # of input figures | 37.0 |
| min # of input figures | 0.0 |
| ave. # of input tables | 4.3 |
| max # of input tables | 53.0 |
| min # of input tables | 0.0 |
| ave. # of reviewers per paper | 3.8 |
| max # of reviewers per paper | 9.0 |
| min # of reviewers per paper | 3.0 |
| ave. # of weaknesses per reviewer | 4.8 |
| max # of weaknesses per reviewer | 39.0 |
| min # of weaknesses per reviewer | 1.0 |
| ave. length of weakness (in words) | 39.1 |
| max length of weakness (in words) | 371.0 |
| min length of weakness (in words) | 2.0 |

# C. More Experiment Results and Details

## C.1. Input Context Scaling Investigation

Figure 4, Figure 5, and Table 10 show the context scaling results of EQINFER, EXPDESIGN, and WEAKNESS.

Table 10: The performance comparison of different input processing methods for  WEAKNESS . We use GPT-4o and GPT-4-Turbo because both accept a maximum of 128k tokens input. We also put the results of AI-SCI in the table for reference. Here, "split-combine" splits the input paper into several pieces, where each piece's length is denoted as "window size"; "no-split" means the conventional input cutting, for example, if the window size is 3,000, then only the first 3,000 words in the paper are used. According to the data statistics, 20,000 words can cover maximum lengths of more than 95% of the papers in our dataset.

| Models | Input Context Processing | Window Size (in words) | S-$F_1$ | S-Precision | S-Recall | ITF-IDF |
|---|---|---|---|---|---|---|
| GPT-4o | split-combine | 3,000 | **47.73** | 42.09 | **55.48** | 5.95 |
| | no-split | 3,000 | 45.74 | **43.45** | 48.54 | 5.92 |
| | no-split | 20,000 | 45.47 | 42.97 | 48.51 | **6.02** |
| AI-SCI | split-combine | 3,000 | **45.05** | 40.02 | **51.91** | 2.23 |
| | no-split | 3,000 | 42.56 | **40.90** | 44.65 | 2.53 |
| | no-split | 20,000 | 42.53 | 40.75 | 44.78 | **2.58** |

## C.2. Human Evaluation on LLM-Generated Novel Experiments

Figure 6 illustrates the evaluation guideline for novel experiments generated by LLMs. We ask 3 senior PhD students to evaluate each paper; that is, if the first two annotators disagree with each other, a third annotator will make a final decision. Table 11 presents several human evaluation cases.

## C.3. Human Evaluation on LLM-Generated Explanation

We ask 5 annotators to evaluate the LLM-generated explanations. Specifically, each of them is assigned 4 or 5 papers, along with the corresponding experiment lists. For each paper, the annotator is given 5 different models' outputs (model names are anonymized), and the annotator has to decide if each LLM-generated explanation is acceptable according to the experiment. We show the human evaluation results in Table 5.

## C.4. Multi-Modal Input Ablation

We post the multi-modal ablation study of EXPDESIGN and WEAKNESS in Table 12 and Table 13.

## D. Data cases and Annotation Platform Illustration

As shown in Figure 8, 9, and 10, we show the sample cases of the three tasks in AAAR-1.0. Meanwhile, we illustrate the screenshot of our annotation platform in Figure 7.

## E. Prompt Templates

In this appendix, we attach all the prompts used in this work, including prompts in data collection and model prediction, as shown in Figure 11, 12, and 13.

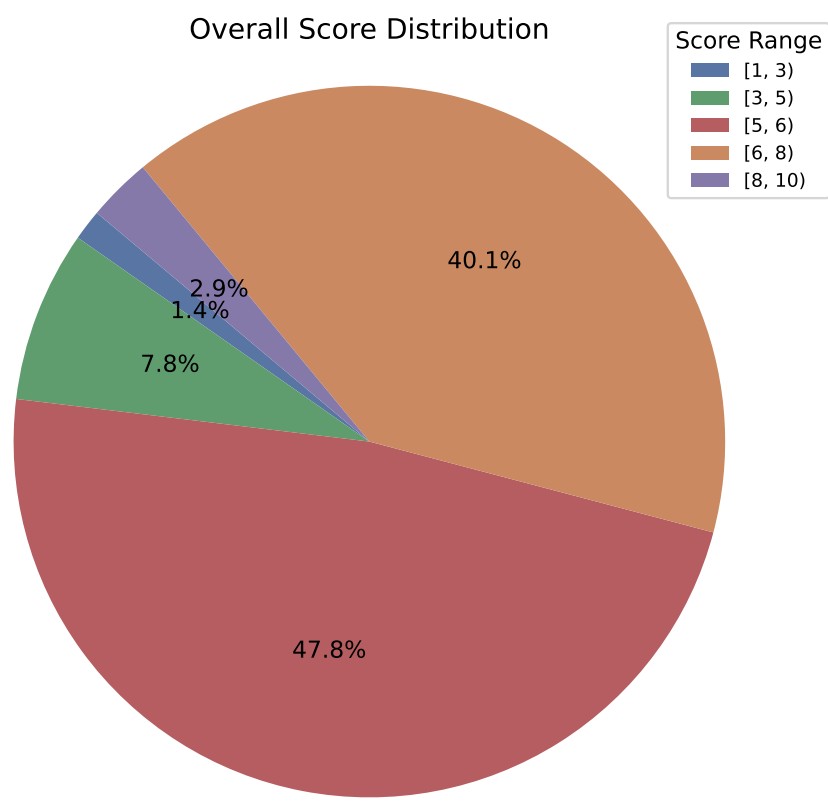

(a) The review score distribution of the papers used in WEAKNESS .

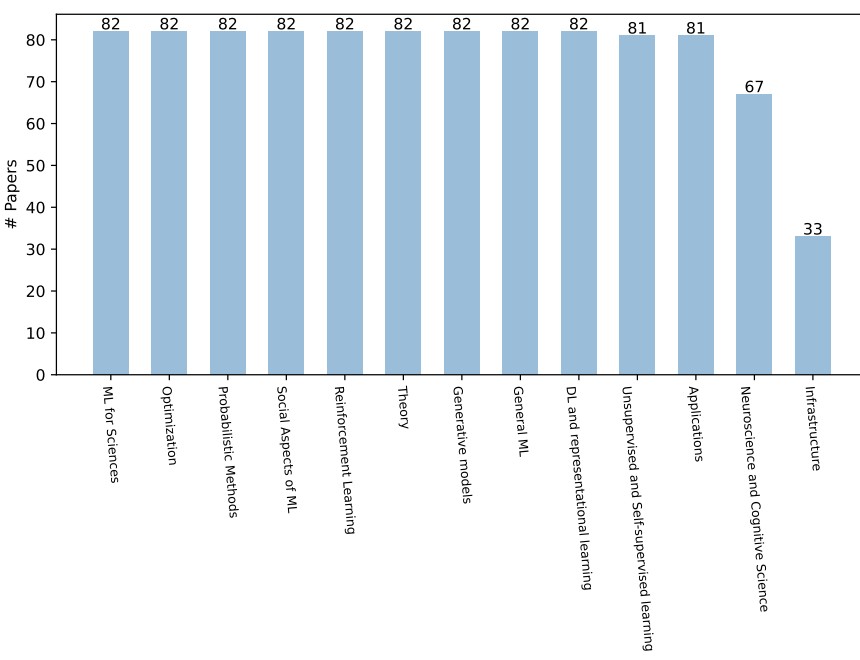

(b) The track distribution of the papers used in WEAKNESS .

Figure 3: The data diversity illustration of WEAKNESS , including the score distribution and track distribution of the papers used in our dataset.

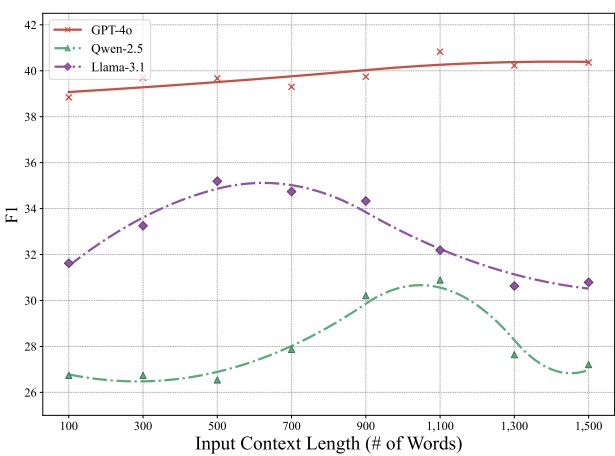

Figure 4: The input context length scaling trend on the EQINFER task.

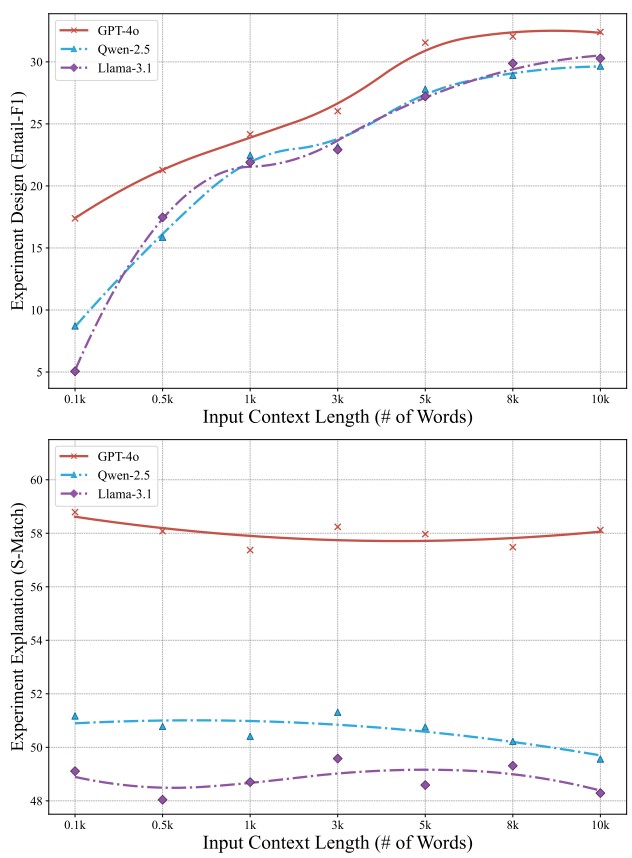

Figure 5: The input context length scaling trend of different LLMs on the EXPDESIGN task.

For each paper, you are given this paper's human-annotated experiments (Column C), along with three different models' prediction experiments (Columns D, G, J)
Those model-generated experiments are all novel experiments that the original human-annotated experiments (Column C) didn't mention. And your task is to evaluate whether these novel experiments are good or not.
Based on the original paper and its experiments, **pls rate the quality of each model-generated experiment.**

**A (necessary experiment)**: Label an experiment with "A" if you think this experiment is **necessary for** this paper.
A "necessary" experiment means if the authors don't include this experiment in the paper, this paper will be highly likely be rejected by the reviewer.
For example, if this paper proposes a novel neural adaptor model, then an ablation study is required to see if having the proposed adaptor can contribute to the performance.

**B (optional experiment)**: label an experiment with "B", if you think this experiment is **an optional choice** for this paper.
For example, if a paper proposes a new metric learning algorithm, conducting a representation space visualization is not required but can be useful for enhancing the explainability of this algorithm.

**C (unrelated experiment)**: label an experiment with "C" if you think this experiment **is unrelated to the core motivation of this paper.** Such as those fancy experiments that we can just omit without any impact.
Note that, if the model-generated experiments are too general, such as simply suggesting an "ablation study" without any details, then you can also categorise it as an unrelated experiment.

In the "Your Assessment" column, write down your assessment of the model-generated experiments,
For example, if there are five novel experiments, write a list with a length of 5: [A, B, C, A, B]
Leave any comments if you are not confident with any of your ratings.

Figure 6: The human guideline for evaluating the LLM-generated novel experiments.

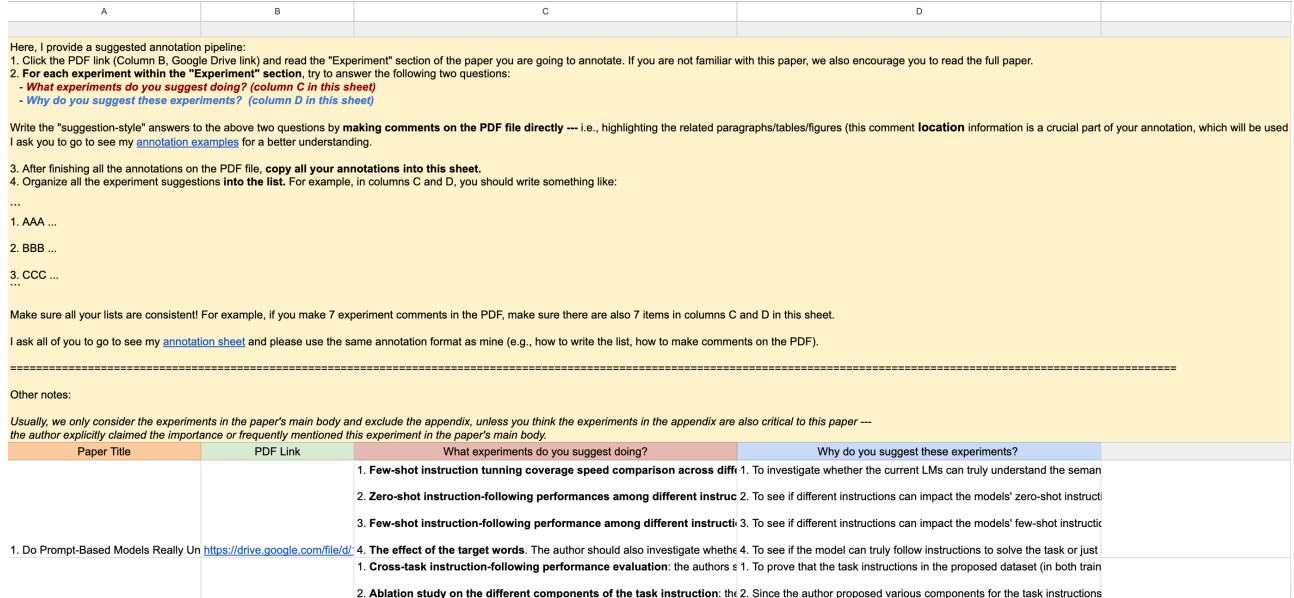

Figure 7: The annotation platform for collecting the annotation of EXPDESIGN. We ask annotators to first make comments on the Google Drive PDF, then move all the annotations to the online Google Doc (for further verification and discussion).

Figure 8: Two sample cases of EQINFER.

Table 11: Examples of human evaluation on the model-generated novel experiments.

| Paper Title | Original Experiments (by human) | Novel Experiment (by LLMs) | Rating |
|---|---|---|---|
| *WiCE: Real-World Entailment for Claims in Wikipedia* | 1. Analysis in Verification Problem Distribution: This paper should provide detailed analysis and statistics about the verification problems in the proposed dataset.

2. Off-the-shelf entailment classification performance: The authors should provide entailment classification performance of existing models on the proposed dataset without fine-tuning.

3. Human Performance: The authors should show human performance on the proposed dataset.

4. Performance of fine-tuned models: The authors should provide the performance of models fine-tuned on the proposed dataset.

5. Performance on the evidence retrieval task: The authors should show the performance on the evidence retrieval task, which is a sub-task of the proposed dataset.

6. Performance of LLMs: The authors should provide the performance of LLMs on the proposed dataset.

7. Retrieval+Entailment: Authors should provide experiments on a framework of retrieving evidence sentences and evaluate entailment by using the retrieved sentences.

8. Analysis of Claim-Split on Downstream Tasks: The authors should analyze how claim-split, the proposed method, is effective on tasks other than the proposed dataset. | Assess model performance on WiCE without fine-tuning to test domain generalization from traditional NLI datasets. | A |
| *MetaMath: Bootstrap Your Own Mathematical Questions for Large Language Models* | 1. Results of multiple LLMs on popular math datasets: The authors should show the performance of multiple LLMs fine-tuned on their dataset on popular math datasets.

2. Performance on open-source models with different sizes: The authors should show the performance of models with different sizes trained on the proposed dataset.

3. Comparison to SOTA closed-source models: The authors show compare the performance of open-source models trained on the proposed dataset and strong close-source models.

4. Evaluate the effect of augmentations: The authors need to perform an ablation study to compare the different argumentation methods they proposed.

5. Analyze Training on Incorrect Answers: The authors should analyze whether wrong answers generated in data augmentation can harm the performance.

6. Evaluate other ways to increase the size of training data: The authors should evaluate other ways to increase the training data size and compare the performance with models trained on their proposed train data.

7. Error Analysis: The authors should analyze the performance of their models in different conditions (e.g., lengths of questions). | Prompt Sensitivity Analysis: Evaluate the sensitivity of MetaMath to different prompt formats or phrasings of mathematical questions. | B |
| *Large Language Models Cannot Self-Correct Reasoning Yet* | 1. Self-Correction with Oracle Labels: The authors should evaluate self-correction performance with oracle labels.
2. Intrinsic Self-Correction: The authors should show performance without using the oracle labels.
3. Analysis of Mistakes in Self-Correction: The authors should analyze the properties of mistakes made in the self-correction framework.
4. Multi-Agent Debate: The authors should evaluate self-correction with multi-agent debate.
5. Prompt Design Analysis: The authors should analyze the influence of prompt design for the initial responses on self-correction performance. | Visualization of learned representations or attention mechanisms to provide insights into the model's inner workings. | C |

Table 12: The figure inputs ablation of EXPDESIGN . For the maximum text input length, same as the setting in Table 2, we use 2,000 and 3,000 words for open- and closed-source models, respectively. For the closed-source GPT-4o and GPT-4, as they have long context window sizes, we use all the figures of each paper. While for InternVL2, we randomly select two figures per input paper.

| Models | Experiment Design | | | Experiment Explanation | | |
|---|---|---|---|---|---|---|
| | En-$F_1$ | En-Precision | En-Recall | S-Match | ROUGE-L | ROUGE-1 |
| GPT-4o | 25.03 | 22.25 | **36.59** | **58.54** | **29.25** | **35.50** |
| w/ figures | **25.39** | **24.35** | 32.80 | 58.53 | 27.87 | 34.30 |
| InternVL2-26B | **24.26** | **39.50** | **14.91** | 50.03 | 29.13 | **34.26** |
| w/ figures | 15.04 | 38.50 | 8.64 | **50.29** | **29.29** | 34.06 |

Table 13: The ablation study about the paper tables and figures of WEAKNESS . Based on the conclusion in Table 10, we use the "split-combine" to process the text input here (2,000 and 3,000 words context window size for open- and closed-source models). For GPT-4o, we use all the table/figure images; while for InternVL2, we randomly select two images per paper, i.e., two random figures, two random tables, or one random figure + table.

| Models | S-$F_1$ | S-Precision | S-Recall | ITF-IDF |
|---|---|---|---|---|
| GPT-4o | **47.73** | **42.09** | **55.48** | **5.95** |
| w/ tables | 46.76 | 41.32 | 54.17 | 5.53 |
| w/ figures | 46.62 | 41.20 | 54.04 | 5.48 |
| w/ tables & figures | 46.58 | 41.17 | 53.98 | 5.36 |
| InternVL2-26B | 41.91 | 41.02 | 43.28 | **1.48** |
| w/ tables | 40.55 | 40.37 | 42.91 | 1.46 |
| w/ figures | **42.88** | **42.10** | **43.76** | 1.46 |
| w/ tables & figures | 42.44 | 42.00 | 43.31 | 1.44 |

| Pre-Experiment Context (Input) | Experiment Design (Output) | Motivation Explanation (Output) |
|---|---|---|
| In this paper, we show that Multilingual BERT (\\mbert{}), released by \\citet{devlin2018bert} as a single language model pre-trained from monolingual corpora in 104 languages, is surprisingly good at zero-shot cross-lingual model transfer, in which task-specific annotations in one language are used to fine-tune the model for evaluation in another language [...] | 1. Expand the definition of overlap. The authors should calculate overlap based on all the words shared between two languages, instead of just shared vocabulary on just the entities.

2. Report performance gains for using some popular language similarity criterion, e.g., WALS.

3. Effect of tokens per word. The authors should perform experiments on more scripts, specifically looking at the effect of words being split into multiple tokens.

4. Control for vocabulary overlap among languages. Choose languages that have large vocabulary overlap and different word order feature. Train on one set of languages and then perform zero shot evaluation on the rest.

5. Ablate the effect of common word pieces by using a non-overlapping tokenizer for different languages. | 1. To check whether non-entity overlap between two languages also contribute to better performance on recognizing the entities. The model may use information from non-entity words to recognize an entity. Additionally, successfully recognizing that a word is not an entity also contributes the performance on the NER task.

2. To understand which features the language model can exploit for cross-lingual transfer. This will give us insights into what typological similarity the multilingual language model can pick up during pretraining.

3. To understand the effect of POS label frequency. The idea is that two languages with similar token to word ratio will result in better cross-lingual transfer. The reason is that continuation tokens should be classified properly and the change in the training corpus of the frequency of continuation tokens will result in different performance.

4. To properly control for the effect of vocabulary overlap. Since large overlap in vocabulary can lead to performance gain, the reported results does not reflect the true impact of word order.

5. To understand the effect of structure of sentences in different languages for cross-lingual understanding of multilingual language models. Since there will be no overlap between different languages the model must learn cross-lingual representations based on syntactic and semantic properties of the languages. |

Figure 9: A sample case of EXPDESIGN .

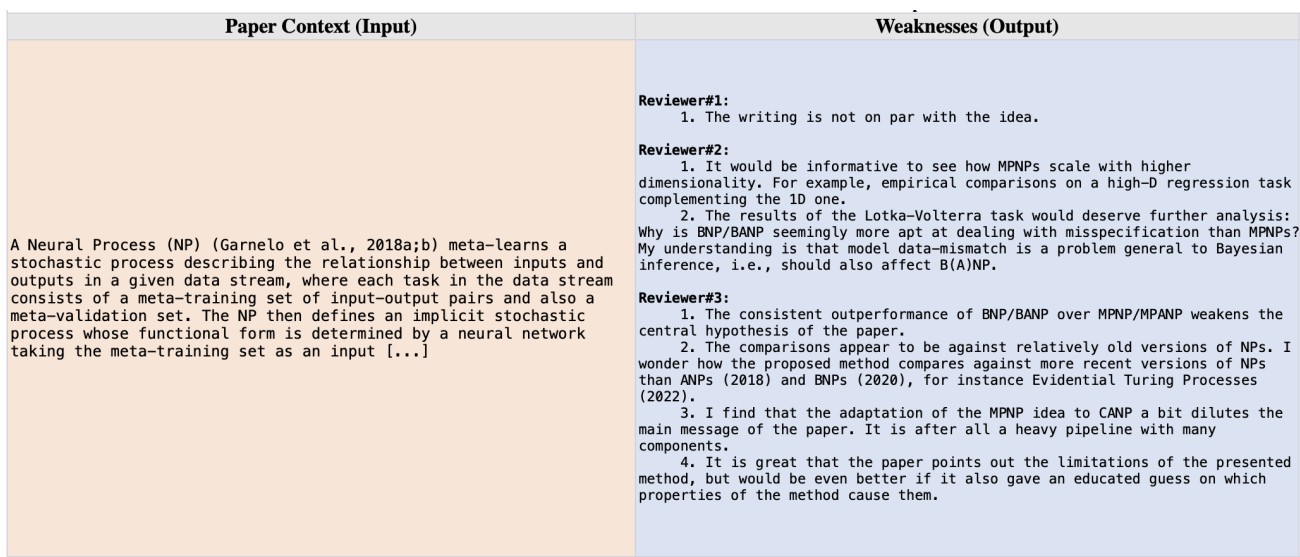

| Paper Context (Input) | Weaknesses (Output) |
|---|---|
| A Neural Process (NP) (Garnelo et al., 2018a;b) meta-learns a stochastic process describing the relationship between inputs and outputs in a given data stream, where each task in the data stream consists of a meta-training set of input-output pairs and also a meta-validation set. The NP then defines an implicit stochastic process whose functional form is determined by a neural network taking the meta-training set as an input [...] | **Reviewer#1:**
    1. The writing is not on par with the idea.

**Reviewer#2:**
    1. It would be informative to see how MPNPs scale with higher dimensionality. For example, empirical comparisons on a high-D regression task complementing the 1D one.
    2. The results of the Lotka-Volterra task would deserve further analysis: Why is BNP/BANP seemingly more apt at dealing with misspecification than MPNPs? My understanding is that model data-mismatch is a problem general to Bayesian inference, i.e., should also affect B(A)NP.

**Reviewer#3:**
    1. The consistent outperformance of BNP/BANP over MPNP/MPANP weakens the central hypothesis of the paper.
    2. The comparisons appear to be against relatively old versions of NPs. I wonder how the proposed method compares against more recent versions of NPs than ANPs (2018) and BNPs (2020), for instance Evidential Turing Processes (2022).
    3. I find that the adaptation of the MPNP idea to CANP a bit dilutes the main message of the paper. It is after all a heavy pipeline with many components.
    4. It is great that the paper points out the limitations of the presented method, but would be even better if it also gave an educated guess on which properties of the method cause them. |

Figure 10: A sample case of WEAKNESS.

| LLM-based Equation Synthesis | LLM-based Equation Filtering | Model Prediction |
|---|---|---|
| ### Task:
You are asked to complete the equation in an NLP paper. Given the context before and after an equation, where the equation is deleted, you should help me recover that equation.

### Requirements:
1. Give me the latex source code of the missed the equation.
2. Only give me the equation, avoid any other explanations.

### Context Before:
```
{The context before the equation.}
```

### Context After:
```
{The context after the equation.}
```

### Equation:
```
{Left part of the ground truth equation}
``` | ### Task:
You are given a source code of a latex equation. Based on your knowledge regarding the Machine Learning and NLP, you should help me identify if this equation has obvious flaw.

### Requirements:
1. If you think this equation has significant flaws, such as grammar errors, logical errors, or any other issues, please mark it as 'Wrong'.
2. Otherwise, please mark it as 'Correct'.
3. Please only give me either 'Correct' or 'Wrong'. Avoid any other explanations.

### Equation:
```
{equation}
```

### Your Answer: | ### Task:
You are given the latex source code of the context before and after an equation in an NLP paper, while this equation is masked. Your task is to identify the correctness of the given candidate equation. Only provide either 'Correct' or 'Wrong'. Avoid any explanations.

### Context Before:
```
{The context before the equation.}
```

### Context After:
```
{The context after the equation.}
```

### Equation:
```
{equation}
```

### Your Answer: |

Figure 11: The prompts used in EQINFER, including both data collection and model prediction.

| LLM-based Leaking Sentence Deletion | Model Prediction (Experiment Design) | Model Prediction (Motivation Explanation) |
|---|---|---|
| You are given a sentence (or a short paragraph) from an ML paper, along with a list of the experiments from this paper; help me decide whether this sentence discusses any experiments in the list.

Let's say, if one sentence includes clues for coming up with any experiments in the list, we call this sentence a 'leaking sentence'; otherwise, if any experiment ideas cannot be inferred from the sentence, we call it a 'non-leak sentence'.

Please give me a '1' if this sentence is a 'leaking sentence'; otherwise, give me a '0'.

### Experiment List:
```
{The experiment list.}
```

### Sentence:
```
{The sentence.}
```

Now, give me your decision (give me either '0' or '1', only the number, without any explanations): | You are partially given an ML paper (in latex), including some useful sections (e.g., 'abstract' and 'introduction') having some basic introductions to the research of this paper, where all the 'experiment' related sections are deleted.

Please first help me carefully read these sections and try to understand the motivations of this research, such as 'what the authors are trying to propose/demonstrate?' and 'what are the main contributions/differences of this paper from others?'

Then, based on your in-depth understanding of this paper, imagine that you are the authors of this paper; what experiments do you have to conduct to prove your research? Namely, you have to **recover the deleted experiments** by providing me with **a list of experiment ideas**, where the list briefly summarizes the experiments the authors should conduct.

Here is an example:
```
{few-shot examples}
```

Here is the target ML paper (partial content):
```
{The context input.}
```

Now, based on this paper, give me a list of experiments the author has to do. Please only give me the list, without any other words.

### Your Experiment List:
``` | You are partially given an NLP paper (in latex), including some useful sections (e.g., 'abstract' and 'introduction') having some basic introductions to this research, where all the 'experiment' related sections are deleted.

Meanwhile, you are also given a list of experiments that try to predict the missed experiments in this paper.

Now, imagine the experiment list you created; you have to explain **why you suggested these experiments**.

Here is an example experiment list:
```
{few-shot examples}
```
Here is the example corresponding explanation list:
```
{few-shot examples}
```

Now, help me look at the following paper:
### Paper:
```
{The context input.}
```

### Experiment List:
```
{The experiment list.}
```

Please give me your explanation list, which should be the same length as the 'Experiment List'; the items of the two lists correspond one-to-one. Only give me the list without any other useless words.
### Explanation List: |

Figure 12: The prompts used in EXPDESIGN, including both data collection and model prediction.

| **Model Prediction (Weaknesses)** |
|---|
| You are given an NLP paper, along with its figure illustrations. Imagine you are a machine learning expert with rich research experience. Please carefully review this paper and identify the weaknesses of this research.

Here is the paper (it might be in partial content):
```
The context input.
```

Now, based on the provided context, give me a list of weaknesses of this research paper (such as '1. XXX\n2. XXX', one point per line).
Note that if the given context is irrelevant to research, such as it is talking about 'acknowledgement', just generate 'No research content'.
Please either give me the weakness list of this research paper or generate 'No research content' to clarify this is not a research paper, without any other words.

### Your Answer: |

Figure 13: The prompts used in WEAKNESS.

