# OpenReview forum: "AAAR-1.0: Assessing AI’s Potential to Assist Research"
_ICML.cc/2025/Conference — ICML 2025 poster_

### Official Review · Reviewer_xpyH · 2025-02-20

**Overall Recommendation:** 4

**Summary:**

This paper introduces AAAR-1.0, a benchmark designed to assess the capabilities of Large Language Models (LLMs) in assisting with research-specific tasks. While most existing benchmarks focus on general-purpose tasks, AAAR-1.0 specifically targets high-level academic reasoning and research assistance, addressing three core challenges:

Equation Inference (EQINFER): Evaluates an LLM’s ability to verify the correctness of equations within research papers, a critical aspect of scientific validation.
Experiment Design (EXPDESIGN): Tests whether LLMs can propose well-structured experimental plans aligned with research objectives.
Paper Weakness Identification (WEAKNESS): Assesses an LLM’s capacity to critically analyze research methodologies and identify key weaknesses.
The dataset is constructed using expert-annotated research examples, ensuring that evaluation aligns with real-world academic challenges. The authors conduct extensive comparative evaluations across a range of LLMs, including GPT-4o, Claude 3.5, Gemini 1.5, Mistral, and Mixtral, revealing:

LLMs struggle significantly with equation reasoning, with performance barely exceeding random baselines.
While LLMs can generate diverse experimental plans, these plans are often misaligned or infeasible for real-world research applications.
In research critique tasks, LLMs can identify broad weaknesses but frequently fail to provide specific, actionable insights comparable to expert reviews.
The paper positions AAAR-1.0 as a necessary step toward evaluating and improving AI’s role in research, highlighting the current limitations and potential future directions for AI-assisted research workflows.

**Claims And Evidence:**

Claim 1: AAAR-1.0 is the first benchmark specifically designed to evaluate LLMs in research-oriented tasks.

While prior work has explored AI-driven research assistance, most datasets focus on code generation, summarization, or retrieval, rather than high-level research tasks.
The authors present AAAR-1.0 as a unique benchmark addressing critical reasoning tasks encountered by academic researchers.
Claim 2: LLMs perform poorly in verifying equations and mathematical reasoning.

Experimental results from the Equation Inference (EQINFER) task show that even the best-performing models (GPT-4o, Claude 3.5) achieve only 46% F1-score, indicating a fundamental gap in symbolic reasoning capabilities.
Open-source models like Mistral and Mixtral perform close to random guessing, reinforcing the difficulty of formal equation verification for LLMs.
Claim 3: LLM-generated experimental designs are often misaligned with real-world research constraints.

In the Experiment Design (EXPDESIGN) task, LLMs produce syntactically correct but practically infeasible experiment proposals.
Human evaluations of 15 model-generated plans show that many experiments were either unnecessary, redundant, or impossible to implement with available resources.
Claim 4: LLMs provide general but shallow critiques in peer review tasks.

In the Paper Weakness Identification (WEAKNESS) task, LLM-generated critiques highlight surface-level issues but often lack depth and specificity compared to human reviewers.
Using an ITF-IDF-based informativeness metric, the paper shows that human-written critiques outperform LLM-generated ones in specificity and actionable feedback.

**Essential References Not Discussed:**

Algorithm of Thoughts: Enhancing Exploration of Ideas in Large Language Models

**Experimental Designs Or Analyses:**

The experiments are controlled and reproducible, ensuring fair model comparisons.
Ablation studies explore the impact of context length, multimodal inputs (figures, tables), and response verification methods.
The finding that multimodal inputs (figures, tables) do not significantly improve performance is valuable, suggesting that LLMs struggle with processing visual research data effectively.

**Methods And Evaluation Criteria:**

The evaluation framework is well-structured, incorporating:

Three benchmark tasks covering distinct aspects of AI-assisted research.
Comparisons across multiple models, including both open-source (Mistral, Mixtral, Qwen, LLaMA) and closed-source (GPT-4o, Claude, Gemini) LLMs.
Task-specific metrics:
F1 Score for equation inference.
Semantic Precision, Recall, and ITF-IDF Informativeness for weakness identification.
Human evaluation for experiment design.

**Other Comments Or Suggestions:**

No

**Other Strengths And Weaknesses:**

Strengths:

Well-structured benchmark tailored to research tasks.
Expert-validated dataset, ensuring high annotation quality.
Provides critical insights into LLMs' limitations in academic reasoning.
Weaknesses:

Limited failure case analysis.
No discussion of computational efficiency.
Evaluation relies heavily on automated metrics instead of human judgments.

**Questions For Authors:**

What are the primary failure patterns in equation inference? Are LLMs failing due to lack of symbolic reasoning or misunderstanding notation?
Would AAAR-1.0 generalize to non-STEM fields like law, medicine, or social sciences?
Could incorporating multi-turn dialogue improve the performance of LLMs in research critique tasks?

**Relation To Broader Scientific Literature:**

This work contributes to ongoing research in AI for research automation, building on:

AI-powered academic assistants (Lu et al., 2024; Si et al., 2024).
Mathematical reasoning in LLMs (Song et al., 2023).
Automated peer review tools (Gao et al., 2024; Liang et al., 2024).

**Theoretical Claims:**

The paper does not propose new theoretical models but provides empirical insights into LLM limitations in symbolic reasoning, research critique, and experiment design.
A discussion on why LLMs fail in equation inference (e.g., limitations in token-level vs. structural mathematical reasoning) would strengthen the study.
A deeper exploration of LLMs’ long-context reasoning capabilities in academic texts could provide additional insights.

---

> ### Author Rebuttal · Authors · 2025-04-01
>
> We sincerely appreciate your detailed review and comments. Below, we provide our comprehensive responses to your questions.
>
> ---
>
>  >Q1. Why LLMs fail on `EQINFER` task & provide the failure patterns discussion.
>
> `EQINFER` leverages a challenging binary inference setting, where LLMs are forced to examine each option separately rather than relying on superficial shortcuts from multiple-choice QA (please refer to the Q1 of Reviewer `qTbh` for more detailed discussion). According to our observations, most LLMs tend to predict any given equation as correct (**a common error pattern for most LLMs**) without performing in-depth reasoning over the paper context. Thus, we assume that the inferior performances of most LLMs result from their **limited long-context reasoning capacity**.
>
> To verify our assumption, we ran the OpenAI o1 model with varying levels of reasoning effort.
>
> |Model   |F1   |
> |--------|-----|
> |o1-low     |42.98|
> |o1-medium  |46.35|
> |o1-high    |**47.12**|
>
> The o1’s performance consistently improves as we increase the reasoning effort. Although this is a simple empirical verification, we believe it highlights the importance of reasoning capacity for this task, particularly for open-source LLMs.
>
> ---
>
> >Q2. Heavy reliance on automatic metrics for evaluation.
>
> Thanks for your suggestions. We proposed task-specific automatic metrics for all the tasks in AAAR to ensure that the public can easily and efficiently reproduce the experimental results reported in our paper.
>
> At the same time, we agree that human judgment is also valuable. As shown in Tables 3 and 4, we conducted a small-scale human evaluation for the model-generated experiment ideas, where we found that despite a few unexpected evaluation results (i.e., false negatives), the results from the automatic metrics generally align well with human judgments, especially when compared with conventional generation metrics like ROUGE. This suggests the reliability of the proposed automatic metrics.
>
> ---
>
> >Q3. Discussion on computational efficiency.
>
> Apologies for missing this detail. When running the open-source LLMs on our local machine, we used [vllm](https://docs.vllm.ai/) to accelerate computational efficiency. Given 4 NVIDIA A100 GPUs for LLM inference, the largest model we utilized in our experiment, Qwen-72B, took approximately 1.25 hours, 0.4 hours, and 1.75 hours for `EQINFER`, `EXPDESIGN`, and `WEAKNESS`, respectively. All the running hyperparameters, such as the maximum model length, can be found in our paper.
>
> In our next version, we will include more details about the computational costs of various LLMs on AAAR.
>
> ---
>
> >Q4. Would AAAR generalize to other fields?
>
> Yes. Our proposed data collection method can ideally be generalized to other disciplines, with AAAR serving as a representative benchmark in the AI/ML field. However, the main constraint is still the requirement for domain experts, as recruiting a reliable and large annotation team is extremely expensive for this kind of research benchmark (see the Q2 discussion of Reviewer `qTbh`).
>
> ---
>
> >Q5. Could incorporating multi-turn dialogue improve the performance of LLMs in research critique tasks?
>
> We assume so. To our knowledge, multi-turn dialogue context can be seen as a structured reasoning path, potentially benefiting LLMs in complex tasks (based on our experience). Meanwhile, some studies have explored dialogue-based collaboration between humans and LLMs or among different LLMs, showing that intra-perspective message sharing can enhance performance in reasoning-intensive tasks [1][2].
>
> ---
>
> ### References:
>
> [1]. [Collaborative Gym: A Framework for Enabling and Evaluating Human-Agent Collaboration.](https://arxiv.org/abs/2412.15701) (*arxiv 2025*)
>
> [2]. [Chain of Agents: Large Language Models Collaborating on Long-Context Tasks.](https://arxiv.org/abs/2406.02818) (*NeurIPS 2024*)

---

### Official Review · Reviewer_aPKE · 2025-03-11

**Overall Recommendation:** 4

**Summary:**

This paper introduces AAAR-1.0, a novel benchmark designed to evaluate the ability of Large Language Models (LLMs) to assist researchers in expert-level tasks. The benchmark comprises three distinct tasks: EquationInference (EQINFER), which assesses the LLM's ability to validate the correctness of equations within a given context; ExperimentDesign (EXPDESIGN), which evaluates the LLM's capacity to design reliable experiments based on a research idea; and PaperWeakness (PAPERWEAKNESS), which tests the LLM's ability to identify weaknesses in research paper drafts. The authors meticulously curated datasets for each task, employing expert annotators to ensure high-quality data. The evaluation process involved a range of both open-source and closed-source LLMs, and the results were analyzed using a combination of quantitative metrics and qualitative assessments. The study found that while LLMs demonstrate some capability in these tasks, their performance is often only slightly above chance, particularly in the EquationInference task, highlighting limitations in their practical utility for advanced research assistance. The paper's core contribution lies in the creation of a benchmark that targets complex, reasoning-intensive tasks that are highly relevant to the research process, moving beyond superficial applications of LLMs.

**Claims And Evidence:**

1. The paper lacks a clear and operational definition of what it means for an LLM to "assist researchers." This ambiguity makes it difficult to interpret the results and understand the practical implications of the benchmark. While the motivation section outlines the challenges researchers face, and the introduction specifies the three tasks, a broader definition is missing. The notion of assistance is too broad and could encompass a wide range of activities, from generating research ideas to writing code, and the paper does not specify which of these are targeted. This lack of clarity makes it challenging to assess the real-world utility of the benchmark.

2. The paper does not provide a clear explanation of how the performance of LLMs on the benchmark translates to their ability to assist researchers in practical scenarios. The connection between the benchmark tasks and real-world research assistance is not well-established, leaving the reader to speculate about the practical utility of the results. For example, it is unclear how well performance on the EquationInference task correlates with the ability of an LLM to help a researcher identify errors in their own equations. While the paper attempts to connect the tasks to real-world scenarios, the explanation is not always explicit and lacks strong supporting evidence or citations.

**Essential References Not Discussed:**

I believe the authors have overlooked discussing a very closely related work: CycleResearcher: Improving Automated Research via Automated Review. In fact, when we compare the data construction of the two papers, it's readily apparent that both AAAR and CycleResearcher's collected Review-5K and Research-14K are oriented towards assisting researchers. For AAAR's ExperimentDesign task, it is actually similar to the "Idea" and "Experiment" sections of Research-14K (and I think they are very similar). For AAAR's PaperWeakness task, it is almost identical to the content of the Review-5K dataset. I understand the authors of AAAR want to focus on benchmarking and evaluation, but the lack of discussion of CycleResearcher, and the absence of using CycleResearcher and CycleReviewer as baseline methods, is difficult to accept.

**Experimental Designs Or Analyses:**

It does not explicitly discuss the limitations of the proposed benchmark. The benchmark not cover all aspects of research assistance, and the tasks biased towards certain types of research or domains. The tasks may also be too narrow, focusing on specific sub-tasks rather than the broader context of research.

**Methods And Evaluation Criteria:**

Convincing enough.

**Other Comments Or Suggestions:**

First and foremost, please, please take the time to clearly define what you mean by "assisting researchers." This isn't just a minor detail; it's fundamental. Make this definition specific and operational – something that truly guides the scope of your benchmark. For instance, you could explicitly state that "assistance" encompasses tasks like identifying errors in equations, suggesting relevant experimental designs, or pinpointing weaknesses in research papers. Ground this definition in the real needs of researchers – what do we actually struggle with? – and let that guide the very design of your benchmark tasks.

It's crucial that you provide a clear and compelling rationale for why you chose these specific tasks. Don't just assume their relevance is obvious. Walk us through your thinking. Discuss the common hurdles researchers face, and explain precisely how your benchmark tasks directly address these challenges. Think about the broader research landscape. Consider how this benchmark could be used across different research domains, and honestly discuss its limitations in those diverse contexts. Transparency here is key.

You need explicitly and honestly discuss the limitations of your proposed benchmark. Don't shy away from this. Analyze potential biases in the tasks. Acknowledge domains not covered. Be upfront about potential biases – is it skewed towards certain research types or domains? Does it truly capture all aspects of research assistance? Discuss the very real possibility of "gaming" – could LLMs be specifically trained to excel on your benchmark without truly understanding the underlying research tasks? This discussion needs to be honest, transparent, and insightful.

**Other Strengths And Weaknesses:**

In my opinion, the true brilliance of this paper absolutely shines through in its introduction of the AAAR-1.0 benchmark. This isn't just another benchmark; it's genuinely novel and undeniably relevant to the booming interest in leveraging LLMs for research support. What I particularly appreciate is how the benchmark tackles expert-level tasks – equation validation, experiment design, and paper weakness identification – this is a remarkably significant contribution. It's refreshing to see a benchmark that moves beyond the shallow, commonplace uses of LLMs and dives into something truly meaningful. Frankly, the authors have brilliantly identified a real gap in existing benchmarks, and they've masterfully filled it with a resource that is precisely what the research community desperately needs.

**Questions For Authors:**

What is the random guess baseline for the PaperWeakness task? how to set?

How do you ensure that the experts involved in the data collection and evaluation process are not biased towards certain LLMs or research domains?

---

Do you know "DeepReview"? I believe this work may be helpful to you.

**Relation To Broader Scientific Literature:**

See in Essential References Not Discussed.

**Theoretical Claims:**

None.

---

> ### Author Rebuttal · Authors · 2025-04-01
>
> Your comments are very much appreciated! We took your comments carefully and tried to address them one by one.
>
> ---
>
> >Q1. The critical references that were missed (i.e., CycleResearch and DeepReview).
>
> Thanks for highlighting these important concurrent works. Specifically, we ran CycleResearch on our benchmark during the rebuttal.
>
> | Method         | S-F1 | ITF-IDF |
> |----------------|------|--------|
> | Llama3.1-70B     |42.78 |2.60     |
> | GPT-4o         |**47.73** |**5.95**    |
> | AI-SCI         |45.05 |2.23    |
> | CycleReview-70B|46.68 |2.65    |
>
> The table above presents the `WEAKNESS` results, where the 70B CycleReview model (based on LLaMA-3.1) achieves an S-F1 score nearly comparable to GPT-4o, highlighting the benefits of post-training for open-source LLMs on this task. However, CycleResearch's ITF-IDF score (the proposed diversity metric) remains similar to LLaMA-3.1 due to a lack of specificity—a common error pattern of LLMs discussed in our paper.
>
> | Method          | En-F1 |
> |-----------------|-------|
> | Llama3.1-70B      | 22.92 |
> | GPT-4o          | **25.03** |
> | CycleResearch-72B| 21.16 |
>
> The table above presents the `EXPDESIGN` results. In our view, CycleResearch may not be a suitable baseline for this task, as it is a policy model designed for whole-paper writing rather than specializing in experiment design. Consequently, applying it to our task requires modifying its original system prompt and task objective, which could make the comparison unfair.
>
> We will provide further observations on CycleResearch in our next manuscript.
>
> ---
>
> >Q2. The definition of “assisting research”.
>
> In terms of 'AI for research', our benchmark differs from existing works in two key aspects.
> - i) **The scope of 'research'**: Since research activities are broad and diverse, we focus on domain-specific, expertise-demanding, and reasoning-intensive tasks that highlight the **irreplaceability of researchers**. For example, writing experimental code is a reasoning-light task, often done by students, whereas determining 'what experiments are necessary' to support a paper’s primary claim is an expertise-demanding task, typically decided by senior advisors. The latter is clearly more challenging and demonstrates that AI models cannot easily replace senior researchers.
> - ii) **'Assisting researchers' rather than 'replacing researchers'**: For high-level research tasks, our benchmark primarily serves an educational purpose — LLMs assist junior researchers by offering imperfect yet insightful ideas rather than governing the entire research process [1]. Relying on LLMs to oversee research and replace human effort compromises academic integrity. For example, we can use LLMs to suggest weaknesses as feedback to help us refine our own manuscripts, rather than directly using model-generated comments for peer review.
>
> ---
>
> >Q3. Reason for choosing the three tasks and their connection with the real-world scenario.
>
> As addressed in Q2, our benchmark focuses on expertise-demanding research tasks that highlight the irreplaceability of researchers. Though more tasks could be included, we prioritize those that are **widely underestimated** in existing works. For example, while writing a paper review is well-studied, identifying a paper’s weaknesses is significantly more challenging than writing a paper summary/strengths [2][3].
>
> `EQINFER` relates to scenarios where LLMs assist in double-checking the correctness of our own equation writing. `EXPDESIGN` mirrors a PhD student seeking a professor’s advice before writing experimental code. `WEAKNESS` represents using LLM feedback to refine our entire research project. Each corresponds to a real-world scenario and strictly aligns with our 'assisting researcher rather than replacing researcher' perspective.
>
> ---
>
> >Q4. The limitation of the benchmark.
>
> Thanks for your suggestion. We agree that discussing limitations would benefit readers and future research. In fact, we initially included a limitation discussion section in our manuscript but removed it to comply with the ICML submission format. Due to the rebuttal word limit, we have provided that section in **[this link](https://anonymous.4open.science/r/ICML2025-rebuttal-A9BF/limitation%20section.png)** and will aim to reintegrate it in future versions.
>
> ---
>
> >Q5. Details about random baseline and how we ensure the data collection is not biased.
>
> For the PaperWeakness task (Table 5), we reported only human performance and did not include a 'random baseline'. Please refer to the Q2 of Reviewer `qTbh` for more details on how we ensured unbiased data collection.
>
> ---
>
> ### References:
>
> [1]. [Collaborative Gym: A Framework for Enabling and Evaluating Human-Agent Collaboration.](https://arxiv.org/abs/2412.15701) (*arxiv 2025*)
>
> [2]. [Can large language models provide useful feedback on research papers? A large-scale empirical analysis.]() (*NEJM AI 2024*)
>
> [3]. [LLMs Assist NLP Researchers: Critique Paper (Meta-)Reviewing.]() (*EMNLP 2024*)

---

> > ### Comment · Reviewer_aPKE · 2025-04-04
> >
> > Thank you to the author for addressing my concerns! I have set my score to 4 to support this work being accepted!

---

### Official Review · Reviewer_WuHG · 2025-03-13

**Overall Recommendation:** 3

**Summary:**

The paper introduces AAAR-1.0, a benchmark for measuring the ability of LLMs to perform 3 key research tasks: mathematical equation understanding, designing experiments, and identifying weaknesses in paper submissions.

The authors curate datasets for each of their chosen research tasks by scraping public research papers and reviews, transforming it, and using expert human annotation to filter and assure quality.

Finally, they evaluate both open and closed-source LLMs on their constructed benchmark and report results along with ablations and analyses.

**Claims And Evidence:**

The main contribution of the paper is their dataset and benchmark, which is well-supported by construction.

Otherwise, there are no significant claims besides their main results which are directly supported by their experiments. I leave specific criticisms of the benchmark and experiments to later sections.

**Essential References Not Discussed:**

None that come to mind.

**Experimental Designs Or Analyses:**

- The choice of unifying context lengths is an unusual one. This arbitrarily limits the performance of models, and adds a lot of complexity to the paper’s discussion and results
  - The paper dedicates many experiments to this topic, which would be relevant for a long-context-focused benchmark, but is only a practical consideration for the AI-research benchmark. (The results are surprising in some cases but it feels mostly like performing hyperparameter sweeps which minimally change the overall conclusions)
  - In my opinion, the benchmark should always directly provide all the necessary information - if models can’t handle it, the models may do worse, but future models will soon have more context length given the quick rate of progress in AI.

**Methods And Evaluation Criteria:**

The overall benchmark is built around a good choice of 3 main tasks, and the authors have done a great job creating interesting datasets for all 3.
- EquationInference - no complaints here, this appears to be a solid and useful dataset for mathematical reasoning with straightforward metrics.
- ExperimentDesign - The dataset is well-constructed. Precision/Recall are suitable metrics. One issue is that you’re making an assumption that the ground truth experiments are exactly what are needed - nothing more, nothing less. If a paper includes an extraneous experiment, or misses a useful experiment that the LLM thinks of, the LLM gets penalized unfairly.
- PaperWeakness - Dataset is well-constructed, and metrics are fine. However, this has the same issue of unfairly penalizing LLMs as I mentioned for ExperimentDesign.

Given the subjective “correctness” of real-world papers/reviews, treating these datasets as ground truths to exactly match, it’s likely that this benchmark is unfair / impossible in some ways. It would be good to have more careful analysis / discussion of what the extent of false positives / false negatives might be in the dataset.

**Other Comments Or Suggestions:**

None.

**Other Strengths And Weaknesses:**

Strengths
- All 3 tasks give really interesting datasets! Even leaving the main benchmark aside, I think they are all really interesting datasets in their own right.
- Very valuable data collection and annotation of papers with expensive experts. Multi-stage annotation with multi-annotation and peer review shows a great care for data quality.
- Interesting results! I was surprised to see the overall weak performance of models on this benchmark, suggesting that it might reveal useful new information about our understanding of models' capabilities.

Weaknesses
- The main concern I have is of data contamination. All the datasets and answers are scraped from publicly available papers on the internet. Before long, models will train on the data and then the benchmark results will no longer be trustworthy. This doesn't seem to be an issue now given the low results, but this will likely limit the lifespan of the benchmark.

**Questions For Authors:**

None.

**Relation To Broader Scientific Literature:**

This work contributes valuable benchmarks and metrics for understanding the ability of LLMs to perform 3 key research tasks that are not currently well-covered by existing datasets. This will be a valuable dataset for future work to build upon. The Related Work in Appendix A does a good job laying out the relevant literature.

**Theoretical Claims:**

No key theoretical claims.

---

> ### Author Rebuttal · Authors · 2025-04-01
>
> Thanks for your efforts in reviewing our manuscript! We're glad you found our dataset interesting, the proposed evaluation metrics reasonable, and the experimental results useful. Below, we address your concerns in detail.
>
> ---
>
> >Q1. Problem for setting the 'ground truth' for `EXPDESIGN` and `WEAKNESS`.
>
> This is a great point. While research is inherently 'open-ended,' human annotations still meet the regular acceptance standards of research — **not a gold standard, but at least a reasonable one**. Therefore, despite its limitations, using human annotations as ground truth remains a practical approach for benchmark construction, as evidenced by recent works that continue to adopt this methodology [1][2].
>
> In this work, to mitigate potential bias from setting human annotations as the ground truth, we establish the evaluation framework that integrates both **automatic metrics** and **human assessment**. For example, in `EXPDESIGN`:
> - First, we use an automatic metric to compute the En-F1 score (Table 2) and identify 'negative' predictions.
> - Then, to *quantify false negative judgments*, experts manually assess the true correctness of these ‘negative’ predictions (Table 3).
>
> To our knowledge, combining automatic metrics with further human assessment represents the 'best' current practice for evaluating research outputs.
>
> ---
>
> >Q2. Analysis for the false positive / false negative cases w.r.t. the 'ground truth'.
>
> Thanks for your helpful suggestion. We have indeed made some analyses in our paper. Specifically, as shown in Table 3, for those model-predicted experiment ideas that do not match the human ground truth, we asked the human experts to evaluate manually.
>
> We found a few model-generated experiment ideas that deviate from the ground truth but are deemed reasonable in manual checks (implying **some false negatives**). Meanwhile, our manual examination confirms that our rigorous peer review process ensures the ground truth's correctness and objectivity (indicating **no notable false positives**).
>
> We hope these results can inspire future work on fairly evaluating research outputs without reliance on ground truth while maintaining efficiency and reproducibility.
>
> ---
>
> >Q3. Concerns on data contamination.
>
> This is an important point to discuss. First, we believe that `EQINFER` and `EXPDESIGN` are less likely to be affected by data contamination, as the ground truth outputs for both tasks have been **reformulated** or **rewritten** from the original source. Output rewriting or distribution perturbation is a widely adopted method to mitigate data contamination [3][4].
>
> However, we acknowledge the potential data leakage issue in the `WEAKNESS` task, as all outputs are directly taken from OpenReview (though we believe this is a common challenge faced by most current benchmark datasets [5]). We maintain that our experimental results still offer valuable insights and can **serve as an upper bound for certain LLMs**, especially if they were pretrained on papers from OpenReview.
>
> Notably, this work introduced AAAR-1.0. Our ongoing efforts involve collecting confidential data that no LLM has touched; we will include that as blind test sets in AAAR-2.0, our next version.
>
> ---
>
> >Q4. Concerns on unifying input length.
>
> Since not all LLMs support long-context reasoning and different models have varying maximum context sizes, we standardized the input length for a fair comparison. We agree that it is crucial to ensure all input information is provided to the LLMs.
>
> To address this, we conducted extensive context length analyses in our manuscript (pls refer to Appendix D1 --- 'Input Context Scaling Investigation'). For example, Figure 4 in the Appendix demonstrates that not all long-context LLMs benefit from the full input information.
>
> ---
>
> ### References:
>
> [1]. [ResearchTown: Simulator of Human Research Community.](https://arxiv.org/abs/2412.17767) (*arxiv 2024*)
>
> [2]. [LLMs Assist NLP Researchers: Critique Paper (Meta-)Reviewing.](https://arxiv.org/pdf/2406.16253) (*EMNLP 2024*)
>
> [3]. [ScienceAgentBench: Toward Rigorous Assessment of Language Agents for Data-Driven Scientific Discovery.](https://arxiv.org/abs/2410.05080) (*ICLR 2025*)
>
> [4]. [ML-Bench: Evaluating Large Language Models and Agents for Machine Learning Tasks on Repository-Level Code.](https://arxiv.org/abs/2311.09835) (*ICLR 2025*)
>
> [5]. [CycleResearcher: Improving Automated Research via Automated Review.](https://arxiv.org/abs/2411.00816) (*ICLR 2025*)

---

### Official Review · Reviewer_qTbh · 2025-03-13

**Overall Recommendation:** 4

**Summary:**

This paper aims to measure the capability of Large Language Models (LLMs) in research-relevant tasks. Specifically, those tasks include 1) Equation Inference, which measures whether the equation is relevant to the given context of the paper, 2) Experiment Design, which measures whether the experimental designs generated by LLMs align with the designs generated by humans, and 3) Paper Weakness, which measures whether the weaknesses of the paper identified by LLMs align with humans. Through extensive experiments, this paper shows that even state-of-the-art LLMs are not sufficient for those advanced research-relevant tasks, and points out that there is room for improvement.

**Claims And Evidence:**

The claims made in the submission are supported by clear and convincing evidence.

**Essential References Not Discussed:**

For the proposed Experiment Design task, indeed there are few recent studies that evaluate the capability of LLMs in generating experiment designs [A, B]. The authors may cite them, and potentially include their approaches in their benchmark evaluation.

[A] ResearchAgent: Iterative Research Idea Generation over Scientific Literature with Large Language Models

[B] Chain of Ideas: Revolutionizing Research Via Novel Idea Development with LLM Agents

**Experimental Designs Or Analyses:**

The evaluation setup for the Equation Inference is not clear. For the given context of the paper, there is one positive equation and three negative equations. As described in Figure 11 (the prompt template), the authors provide those four options as well as the given context, and prompt the model to predict one that is the relevant equation for the given context. If so (i.e., if the setting is the multiple-choice question answering), it is not clear how to formulate the "All-Positive baseline" that predicts all equations as positive. Also, as shown in the results of Table 1, most LLMs tend to predict the equation as positive, and there are no substantial improvements over the "All-Positive baseline". In this regard, I am wondering whether the LLMs still do not show substantial performance improvement over the "All-Positive baseline" if the authors use the multiple-choice question-answering setup.

**Methods And Evaluation Criteria:**

The proposed methods and evaluation criteria are mostly reasonable. However, one critical concern about the overall quality and reliability of the proposed dataset collection process and evaluation setup is that the authors seem to be annotating data with five (or a few) PhD students. While the authors claim that they are senior researchers, I view them as still students, and it may be beneficial to check the quality of the collected data and the annotated experimental results with more seasoned researchers. Also, according to this, I think the authors may need to tone down their claim of performing annotations with senior researchers. Lastly, in addition to them, it is questionable how they were recruited, how much they were compensated, and how diverse they are across domains. Providing this information seems conventional for benchmark work.

**Other Comments Or Suggestions:**

It would be interesting to see the performance where LLMs are prompted with the few-shot examples to perform the given task. The LLMs might not be familiar with the given tasks (as they are a bit different from a typical suite of evaluation benchmarks); however, by learning from examples in-context, they can capture the core principle of the given task and then achieve significantly higher performance.

**Other Strengths And Weaknesses:**

Please see my previous comments.

**Questions For Authors:**

I feel like this paper is borderline, and I would like to increase my score if the authors would address all my comments.

**Relation To Broader Scientific Literature:**

There is a growing body of literature on using AI for science (to accelerate it), and this paper is relevant to this topic, which is very important and timely.

**Theoretical Claims:**

N/A

---

> ### Author Rebuttal · Authors · 2025-04-01
>
> Many thanks for your detailed and comprehensive review. We're pleased you found our evaluation criterion reasonable and the literature topic important. As shown below, we address your main concerns one by one:
>
> ---
> > Q1:  Unclear evaluation setup for the Equation Inference.
>
> We apologize for the confusion. Although `EQINFER` was originally designed for a MCQA setting (which explains the 1:3 positive-to-negative distribution), we ultimately adopted a **binary inference setup**, as shown in Figure 1 / Footnote 1. This change was made because of two reasons:
> - i) We found that the MCQA setup contains shortcuts --- LLMs may focus on superficial character differences among the options rather than reasoning through the paper context. In contrast, binary inference requires LLMs to evaluate the correctness of **each option separately**, eliminating any straightforward shortcuts.
> - ii) In real-world equation reviewing during paper submissions, the task also involves determining whether the author-written equation is valid based on the context. We are making our 'AI assists researcher' **setting more realistic**.
>
> If you're interested, you can find the MCQA setup performances at this **[link](https://anonymous.4open.science/r/ICML2025-rebuttal-A9BF/EQINFER_MCQA_performance.png)**. As shown, LLMs generally achieve higher scores due to the MCQA shortcut. We apologize again for the misleading MCQA prompt template in the appendix and will update it in our next manuscript.
>
> ---
>
> > Q2. The reliability of the employed annotators.
>
> Thank you for pointing out this critical argument. We will include the following recruitment details in our next version:
> - **Recruiting**: we posted an online recruiting form with strict qualification requirements: i) more than 4 years of AI research experience; ii) more than 5 publications, with at least one first-authored publication in leading AI venues; iii) reviewed 10+ peer publications. The form was then shared through social media.
> - **Annotator Profile**: over 10 annotators were selected, all from renowned academic institutions with strong ML/AI research backgrounds (**not just five students**), including some professors.
> - **Domains**: we aimed to include annotators with diverse AI expertise, including NLP, CV, and ML theory.
> - **Reward**: given the challenge of collecting expertise-level data, we offer a high payment of \\$70 per hour to all annotators. Additionally, they receive a \\$5 bonus for each low-quality sample identified or valid response made during peer discussions.
>
> We will follow your suggestion to tone down the term 'senior researcher' in the next version. It is important to emphasize that we have made every effort to ensure all annotators have a strong background and are fully engaged in rigorous data collection and examination, which was also acknowledged by Reviewer `WuHG`.
>
> ---
>
> > Q3. Performance of adding few-shot examples.
>
> This is a good suggestion. Here, we provide the performance of adopting the few-shot examples.
>
> | Model         | 0-shot | 2-shot |
> |---------------|--------|--------|
> | Mistral-7B    | 28.45  | 30.45  |
> | Qwen-72B      | 31.22  | 33.09  |
> | GPT-4o        | 40.35  | 40.61  |
> | o1            | 46.35  | 46.28  |
>
> The above table illustrates the results of `EQINFER`, we found that adopting few-shot examples only improves the F1 score of smaller open-source LLMs. We believe the examples mainly serve as format guidance for classification in this case, while the reasoning ability of LLMs plays a more critical role in EQINFER (see our Q1 discussion with Reviewer `xpyH`).
>
> | Model       | 0-shot  | 2-shot  |
> |-------------|-----|-----|
> | Mistral-7B  |1.17 |0.89 |
> | Qwen-72B    |1.21 |0.94 |
> | GPT-4o      |5.95 |5.02 |
> | o1          |5.63 |4.78 |
>
> The above table illustrates the results of `WEAKNESS`, we found that few-shot examples even negatively impact the ITF-IDF scores of various LLMs. Our observation suggests that adding examples restricts LLMs' creativity in generating novel weaknesses. For instance, if we include 'lack of novelty' as an example weakness, LLMs tend to repeat this weakness across different papers.
>
> For `EXPDESIGN`, due to the list format generation, this is the only task where we observed significant and consistent performance improvement after adopting few-shot examples (which we’ve already used in our paper, as seen in Figure 12).
>
> ---
>
> > Q4. Missing references.
>
> Thanks for suggesting highly relevant references. We will include a discussion on them in our next version. Additionally, we have conducted further baseline experiments to make our evaluation more comprehensive (please see our Q1 discussion with Reviewer `aPKE`).
>
> ---

---

> > ### Comment · Reviewer_qTbh · 2025-04-04
> >
> > Thank you for your detailed response, which effectively addresses all the concerns I raised. I increase my score to 4 (accept) from 2.

---

### Decision · Program_Chairs · 2025-05-01

**Decision:**

Accept (poster)

**Comment:**

This paper introduces AAAR-1.0, a benchmark designed to evaluate LLMs' capabilities in assisting with research-oriented tasks across three dimensions: validating equation correctness, designing experiments, and identifying paper weaknesses. All four reviewers recommended acceptance, appreciating the paper's novel contribution in addressing expertise-intensive research tasks, high-quality dataset with rigorous expert annotation, comprehensive evaluation across various LLMs, and valuable insights into current limitations. During the rebuttal, the authors effectively addressed concerns about the evaluation setup, annotator reliability, baseline comparisons, and provided a clearer definition of "assisting research." The authors' responses were thorough and satisfactory, leading some reviewers to increase their ratings. Given the paper's strong contributions, thorough evaluation, and comprehensive responses to reviewer feedback, I recommend accepting this paper for ICML 2025 as it provides valuable insights into LLMs' capabilities in supporting research activities.